# MLR and dMLR Predict Locoregional Control and Progression-Free Survival in Unresectable NSCLC Stage III Patients: Results from the Austrian Radio-Oncological Lung Cancer Study Association Registry (ALLSTAR)

**DOI:** 10.3390/jcm14248876

**Published:** 2025-12-15

**Authors:** Alexandra Hochreiter, Markus Stana, Marisa Klebermass, Elvis Ruznic, Brane Grambozov, Josef Karner, Martin Heilmann, Danijela Minasch, Ayurzana Purevdorj, Georg Gruber, Raphaela Moosbrugger, Falk Röder, Franz Zehentmayr

**Affiliations:** 1Department of Radiation Oncology, Paracelsus Medical University, 5020 Salzburg, Austria; a.hochreiter@salk.at (A.H.); m.stana@salk.at (M.S.); e.ruznic@salk.at (E.R.); b.grambozov@salk.at (B.G.); j.karner@salk.at (J.K.); f.roeder@salk.at (F.R.); 2Klinikum Ottakring, 1160 Vienna, Austria; marisa.klebermass@gesundheitsverbund.at; 3Department of Radiation Oncology, Comprehensive Cancer Centre, Medical University Vienna, 1090 Vienna, Austria; martin.heilmann@akhwien.at; 4Department of Radiation Oncology, Comprehensive Cancer Centre, Medical University Innsbruck, 6020 Innsbruck, Austria; danijela.minasch@tirol-kliniken.at; 5Klinikum Hietzing-Rosenhügel, 1130 Vienna, Austria; 6Ordensklinikum, Department of Radiation Oncology, Johannes Kepler University, 4020 Linz, Austria; georg.gruber@ordensklinikum.at; 7Department of Pulmonology, Paracelsus Medical University, 5020 Salzburg, Austria; r.moosbrugger@salk.at; 8Institute of Research and Development of Advanced Radiation Technologies (radART), Paracelsus Medical University, 5020 Salzburg, Austria

**Keywords:** non-small cell lung cancer, RWD, durvalumab, ALLSTAR, blood biomarkers, MLR, dMLR

## Abstract

**Background**: As demonstrated by the PACIFIC trial, biomarker-driven patient selection is crucial. While treatment based on programmed death ligand-1 (PD-L1) and mutational status have become routine, tests for biomarkers available from pretherapeutic blood samples are currently a topic of scientific interest. **Methods**: This analysis was conducted on patients from the ALLSTAR RWD study, which is a nationwide, prospective registry for inoperable non-small cell lung cancer (NSCLC) stage III. Patients were amenable if they had a full routine pre-treatment blood sample, from which the following biomarkers were extracted: neutrophil-to-lymphocyte ratio (NLR), derived neutrophil-to-lymphocyte ratio (dNLR), platelet-to-lymphocyte ratio (PLR), monocyte-to-lymphocyte ratio (MLR), derived monocyte-to-lymphocyte ratio (dMLR) and lactate dehydrogenase (LDH) levels. The intention was to find a cutoff for each of these biomarkers to predict locoregional control (LRC), progression-free survival (PFS) and overall survival (OS). **Results**: MLR and dMLR demonstrated their predictive potential with cutoff values of 0.665 and 0.945, respectively. Stratifying the whole cohort by means of these cutoffs demonstrated significantly better locoregional control for patients below the threshold, both in the whole cohort (N = 175; 55.7% vs. 75.5%; *p*-value = 0.018) and in the Durvalumab subgroup (N = 106; 57.5% vs. 77.3%; *p*-value = 0.030). Similar findings were observed for PFS in the whole cohort (N = 175; 20.5% vs. 56.1%; *p*-value *p* < 0.001) and in the Durvalumab subgroup (N = 106; 31.2% vs. 64.6%, *p*-value < 0.001). dMLR could also significantly predict PFS (N = 173; 17.4% vs. 56.3%; *p*-value < 0.001), which was corroborated in the Durvalumab subgroup (N = 108; 23.1% vs. 64.1%; *p*-value = 0.003). **Conclusions**: This explorative analysis demonstrates the predictive potential of MLR and dMLR for LRC and PFS. These blood biomarkers can be readily integrated into clinical routines since they are easily available.

## 1. Introduction

Accounting for 18% of cancer-related deaths worldwide, lung cancer is one of the most commonly diagnosed malignancies [1]. Of these, 70% are diagnosed as non-small cell lung cancer (NSCLC), with one third already at a locally advanced UICC stage IIIa to IIIc [1].

Concurrent chemoradiotherapy followed by consolidation with the PD-L1 inhibitor Durvalumab represents the current standard of care for patients with locally advanced, unresectable stage III NSCLC. While this treatment paradigm, established by the PACIFIC landmark trial, has drastically improved survival outcomes, with 5-year overall survival (OS) and progression-free survival (PFS) rates of 42.9% and 33.1%, respectively [2,3], locoregional control (LRC), with approximately 70% at two years [4], has remained in the same range as in the historical series [5,6,7,8,9]. Overall, the prognosis in this heterogeneous patient collective may remain dismal, with only a subset of approximately 20% of the patients achieving long-term benefit from immune checkpoint inhibition (ICI) treatment [10]. Consequently, biomarker-driven patient selection is essential in order to appropriately adjust treatment strategies.

In the past decade, PD-L1 has gained a prominent role, since patients—at least in Europe—are only amenable to Durvalumab therapy based on their PD-L1 status [11], although PD-L1-negative patients also respond well to immunotherapy [2,3]. Additionally, pretherapeutic testing for a panel of druggable targets, such as EGFR, KRAS, ALK/ROS and others, has become standard worldwide [12]. These tests—although necessary—are time-consuming, cumbersome and expensive.

In contrast, blood-based biomarkers that are produced by inflammatory processes can easily be taken from routine pretherapeutic blood samples and are therefore fast and cost-effective [10,13,14,15,16,17,18,19,20,21]. Hence, they have emerged as potential prognostic tools in various malignancies, since they reflect the balance between pro-tumorigenic inflammation and anti-tumour immune response. In particular, high levels of neutrophils and monocytes may foster an immunosuppressive microenvironment and release pro-angiogenetic factors, thereby promoting tumour growth in the wake of inflammatory remodelling of the peritumoral tissue [22]. On the other hand, lymphocytes, which are part of the human cellular immunity, are involved in anti-tumour response by inhibiting tumour cell proliferation [23,24]. In regard to NSCLC, the following markers have been investigated thus far: the neutrophil-to-lymphocyte ratio (NLR), derived NLR, the lymphocyte-to-monocyte ratio (LMR), the monocyte-to-lymphocyte ratio (MLR) and the platelet-to-lymphocyte ratio (PLR) [13,14,16,17,25]. Moreover, various studies have demonstrated that a lower LMR (conversely, a higher MLR) is associated with a better PFS and OS [13,15,16,17]. The role of dMLR was first evaluated in the PAC-KR trial, showing that a higher dMLR is a predictor of severe radiation pneumonitis, but no survival analyses were conducted [13]. Thus far, the current study is the first to investigate the predictive/prognostic significance of dMLR in time-to-event analyses.

The objective of the present exploratory study, conducted within the nationwide prospective ALLSTAR registry, was therefore to evaluate the prognostic significance of inflammatory blood biomarkers, particularly MLR and dMLR, in unresectable stage III NSCLC patients in regard to LRC, PFS and OS.

## 2. Materials and Methods

### 2.1. Patients

Patients included in the current analysis were treated between 21 March 2020 and 11 April 2023. Data were collected within the ALLSTAR project, which is a nationwide, prospective registry for inoperable NSCLC UICC stage III patients. All patients provided written informed consent. Approval by the ethics committee of the federal state of Salzburg was obtained on 20 March 2020 (approval number: 1002/2019). At each centre, the local multidisciplinary tumour board was responsible for treatment decisions. Patients aged 18 years or older with histologically or cytologically confirmed unresectable stage III NSCLC (TNM Version 8) who were treated with definitive chemoradiotherapy with or without ICI were included. Details on inclusion criteria and follow-up procedures are described elsewhere [26]. All analyses were conducted first in the whole cohort (N = 183) and in a second step in a subpopulation of these patients who were treated with Durvalumab (N = 112).

### 2.2. Chemoradioimmunotherapy

Radiation treatment was administered according to clinical practices at each centre, preferably with advanced technologies such as intensity-modulated radiotherapy (IMRT), volumetric arc therapy (VMAT) or 3D radiation. As for the total radiation dose, 60–66 Gy in 2 Gy daily fractions was considered as the standard of care. To ensure comparability, total irradiation doses were recalculated as biologically equivalent doses in 2 Gy fractions ${(EQD}_{2})$ using the following formula:(1)$${EQD}_{2}=\frac{d+\frac{\alpha }{\beta }}{2+\frac{\alpha }{\beta }}$$
with d for the single dose, D for the total dose and $\frac{\alpha }{\beta }$ set at 10 for tumour tissue.

Although concomitant chemoradiotherapy is the international standard of care [5,6,7,8,9,27], in ALLSTAR, both sequential and concomitant modes—in accordance with local practices at each centre—were allowed. Chemotherapeutic agents were carboplatinum or cisplatinum combined with pemetrexed, taxane, gemcitabine or vinorelbine, depending on histology.

### 2.3. Endpoints and Statistics

The primary endpoints of the study were locoregional control (LRC), progression-free survival (PFS) and overall survival (OS). LRC was defined as freedom from intrathoracic tumour progression (within or adjacent to the radiotherapy field, including regional nodes), according to the definition of Machtay [28]. PFS was defined as the period between diagnosis and tumour progression at any site. Time-to-event analyses were calculated using the Kaplan–Meier method with the day of pathological (either histological or cytological) diagnosis as the index date. Log-rank testing was used for subgroup comparisons. Multivariate analyses were performed with the Cox proportional hazard regression. MLR and dMLR were tested together with either baseline characteristics (age, sex, histology, UICC) or treatment parameters (GTV_Tumour_, GTV_Lymphnodes_, EQD_2Tumour_, EQD_2Lymphnodes_, chemoradiotherapy sequence, ICI treatment). Because of potential collinearity, MLR and dMLR were analysed in separate models. A *p*-value of <0.05 was considered statistically significant. The Benjamini–Hochberg method was used to adjust for multiple testing.

### 2.4. Blood Biomarker Calculation

This explorative study included only patients with pre-treatment blood samples, which were used to extract the following biomarkers: the neutrophil-to-lymphocyte ratio (NLR), derived neutrophil-to-lymphocyte ratio (dNLR), platelet-to-lymphocyte ratio (PLR), monocyte-to-lymphocyte ratio (MLR), derived monocyte-to-lymphocyte ratio (dMLR) and lactate dehydrogenase (LDH) levels. dNLR was calculated as follows:(2)$$\frac{absolute\;neutrophil\;count}{white\;blood\;cell\;count-absolute\;neutrophil\;count}$$

For dMLR the following formula was used:(3)$$\frac{white\;blood\;cell\;count-absolute\;neutrophil\;count-absolute\;lymphocyte\;count}{absolute\;lymphocyte\;count}$$
[13].

The programming language Python version 3.10.4 [29] was used to identify thresholds with predictive value for the mentioned clinical endpoints. Every single possible combination of two groups (one with its members below and the other one above a specific value) within the cohort was tested. For each combination, log-rank *p*-values were calculated and plotted against the cutoff value. If more than one consecutive *p*-value < 0.05 was detected, this was considered a robust signal. The “optimal” cutoffs had to fulfil the following criteria: (1) They had to lie within the range of published values with a +/−25% margin. (2) They should be valid for as many clinical endpoints as possible in both the whole cohort and the Durvalumab subgroup. Please find the study overview in Appendix A.

## 3. Results

### 3.1. Patient and Treatment Characteristics

Patients from the ALLSTAR registry [26,30] with at least one clinical follow-up and a standard pre-treatment blood sample were included in the study, which resulted in a total of 183 patients eligible for the current analysis. A total of 111 (60.7%) patients were male and 72 (39.3%) were female, with a median age of 67.3 years (range 36.4–90.7). The majority (93.4%) had a good ECOG performance status of 0–1. All patients had histologically verified NSCLC stage III. Slightly more than half of the patients (51.9%) had tumours classified as adenocarcinoma, 42.1% as squamous cell carcinoma and 6.0% as not otherwise specified. More than two-thirds (69.9%) were PD-L1-positive, i.e., >1%. Most patients presented with T3 and T4 tumours (31.1% and 41.6%), and 61.2% (*n* = 112) had positive N2-status. UICC stages IIIa, IIIb and IIIc were present in 34.9%, 45.4% and 19.7% of patients, respectively. Chemoradiotherapy was administered sequentially in 126/183 patients (68.9%). The preferred radiation treatment technique was VMAT/IMRT in 97.3% of patients, with an average total irradiation dose to the gross tumour volume (GTV) of 65 Gy EQD2. The patient and treatment characteristics are summarized in Table 1 and Table 2.

A total of 112 (61.2%) patients received maintenance immunotherapy with Durvalumab. In this subgroup, 70 patients (62.5%) were male and 42 were female (37.5%). The median age was 67.4 years (range 40.6–83.9). An ECOG performance status of 0–1 was observed in 94.6% of patients. Furthermore, 61 patients (54.5%) presented with adenocarcinoma, whereas 47 (42.0%) had squamous cell carcinoma. PD-L1 status > 1% was found in the majority of patients (*n* = 93; 83.0%). Similar to the whole cohort, the majority of patients in the Durvalumab subgroup had T3 (33.9%) and T4 (36.6%) tumours with positive N2 (67.9%) or N3 (18.7%) lymph nodes. UICC stages IIIa, IIIb and IIIc were distributed as follows: 36.6%, 46.4% and 17.0%. Again, the majority of patients (81/112; 72.3%) received chemoradiotherapy sequentially. Again, in this subgroup, the most commonly applied radiation treatment technique was VMAT/IMRT (97.3% of patients), with a median total EQD2 for GTVTumour of 67.1 Gy. For details see Table 1 and Table 2.

### 3.2. Blood Biomarkers

Following an initial exploratory analysis of NLR, MLR, PLR, dNLR, dMLR and LDH, subsequent analyses focused primarily on the most conclusive outcomes in terms of the criteria defined at the end of Section 2.4. (Appendix A). Only MLR and dMLR, which yielded the strongest and most consistent signal in the Python analysis (Appendix A), were therefore investigated further. The range of published LMR values (reviewed by Chan et al. [16]) was between 2.50 and 5.09, with a specific threshold of 1.8 mentioned by Chen [14]. As we calculated MLR, which is the reverse of these published values, this 1.8 threshold would correspond to 0.556. For dMLR no published data were available. In summary, the optimal cutoff values for MLR and dMLR valid for both LRC and PFS were 0.665 and 0.945, respectively.

### 3.3. Locoregional Control, Progressionfree Survival, Overall Survival

With a median follow-up of 30.9 months (95%-CI: 27.7–33.9), the 2-year LRC was 72.3% in the whole cohort (N = 183, Figure 1a). The comparison between patients stratified by MLR cutoff revealed a significantly poorer LRC for patients above the threshold of 0.665 compared to those below (N = 175; 2-year LRC rates of 55.7% versus 75.5%; log-rank *p*-value = 0.018; Figure 2a). Similar findings were observed in the Durvalumab subgroup, where a higher MLR was also associated with worse LRC (N = 106; 2-year LRC rates of 57.5% vs. 77.3%; log-rank *p*-value = 0.030; Figure 2b). Additionally, patients with a dMLR >0.945 trended towards a worse LRC compared to those below the threshold, with 2-year LRC rates of 58.1% versus 75.6% (N = 173; log-rank *p*-value = 0.054; Figure 2c). Although not statistically significant, this finding was corroborated in the Durvalumab subgroup (N = 108; 2-year LRC rates of 64.5% versus 76.3% Figure 2d).

The overall 2-year PFS was 48.6% (Figure 1b). Patients with an MLR above the threshold of >0.665 had a significantly poorer 2-year PFS compared to those below (N = 175; 2-year PFS rates 20.5% vs. 56.1%; log-rank *p*-value *p* < 0.001; Figure 3a). Similar results were observed in the Durvalumab subgroup, with a 2-year PFS of 31.2% versus 64.6%, favouring patients with a lower MLR (N = 106; log-rank *p*-value < 0.004; Figure 3b). As for the dMLR, patients above the threshold (>0.945) had a significantly worse 2-year PFS of 17.4% versus 56.3% (N = 173; log-rank *p*-value < 0.001; Figure 3c). Durvalumab patients with a higher dMLR (>0.945) showed a significantly poorer 2-year PFS of 23.1% versus 64.1% (N = 108; log-rank *p*-value = 0.003; Figure 3d).

As for OS, we could neither find a threshold for MLR nor for dMLR that was able to predict outcomes.

### 3.4. Multivariate Analyses of Biomarkers for Locoregional Control and Progression-Free Survival

When the MLR was tested in the univariate (UVA) and multivariate models (MVA) together with baseline and treatment characteristics, the following variables had a significant impact on the LRC: histology (*p*-value = 0.004; corrected *p*-value = 0.02; HR 2.6; 95% CI 1.35–4.89; Table 3a) and MLR (*p*-value = 0.012; corrected *p*-value = 0.070; HR 0.38; 95% CI 0.18–0.81; Table 3b). This finding was corroborated in the Durvalumab subgroup (histology: *p*-value < 0.01; corrected *p*-value 0.047; HR 3.2; 95% CI 1.3–7.7; Table 3a; MLR: *p*-value < 0.05; corrected *p*-value = 0.15; HR 0.4; 95% CI 0.16–0.998, Table 3b). Additionally, GTV_Tumour_ was also a significant factor with respect to the LRC in the Durvalumab cohort (*p*-value = 0.01; corrected *p*-value = 0.07; HR = 1.0095; 95% CI 1.0022–1.017). A separate analysis with the same baseline and treatment characteristics combined with the dMLR instead of MLR revealed that histology was the only predictive parameter for LRC, both in the whole cohort (*p*-value < 0.003; corrected *p*-value = 0.013; HR 2.75; 95% CI 1.4–5.3; Table 3c) and in the Durvalumab subgroup (*p*-value = 0.007; corrected *p*-value = 0.03; HR 3.3; 95% CI 1.4–7.9; Table 3c). Furthermore, the dMLR had an impact on LRC in the whole cohort (*p*-value = 0.019; corrected *p*-value = 0.11; HR = 0.396; 95% CI 0.18–0.86; Table 3d).

When MLR was tested in the MVA model together with baseline and treatment characteristics, the following variables had a significant impact on PFS: UICC (*p*-value = 0.004; corrected *p*-value < 0.01; HR 2.1; 95% CI 1.28–3.56; Table 4a), MLR (Table 4a: *p*-value < 0.001; corrected *p*-value = 0.004; HR 0.43; 95% CI 0.26–0.7; Table 4b: *p*-value < 0.001; corrected *p*-value < 0.005; HR 0.39; 95% CI 0.22–0.68) and EQD2_Tumour_ (*p*-value < 0.03; corrected *p*-value = 0.08; HR 0.97; 95% CI 0.95–0.9973; Table 4b). In the Durvalumab subgroup, risk factors for a PFS were histology (*p*-value = 0.044; corrected *p*-value = 0.073; HR 1.91; 95% CI 1.03–4.3; Table 4a), UICC (*p*-value = 0.04; corrected *p*-value = 0.073; HR 1.91; 95% CI 1.02–3.6; Table 4a) and MLR (Table 4a: *p*-value = 0.014; corrected *p*-value = 0.073; HR 0.4; 95% CI 0.22–0.85; Table 4b: *p*-value = 0.0059; corrected *p*-value 0.035; HR 0.365; 95% CI 0.178–0.75). When repeating the same analysis with dMLR instead of MLR, UICC (*p*-value < 0.009; corrected *p*-value = 0.023; HR 1.96; 95% CI 1.18–3.2; Table 4c) and dMLR (Table 4c: *p*-value = 0.04; corrected *p*-value = 0.02; HR 0.45; 95% CI 0.26–0.78; Table 4d: *p*-value = 0.002; corrected *p*-value = 0.014; HR 0.41; 95% CI: 0.2296–0.7264) were significant factors associated with PFS in the whole cohort. This result was also found in the Durvalumab subgroup: UICC (*p*-value = 0.048; corrected *p*-value 0.12; HR 2.01; 95% CI 1.006–4.0, Table 4c) and dMLR (Table 4c: *p*-value = 0.018; corrected *p*-value = 0.088; HR 0.4; 95% CI 0.22–0.86; Table 4d: *p*-value = 0.0017; corrected *p*-value = 0.01; HR 0.314; 95% CI 0.15–0.65).

## 4. Discussion

This analysis from the nationwide ALLSTAR RWD study demonstrates that the MLR and dMLR predict outcomes in inoperable NSCLC UICC stage III. Patients with biomarker values below the thresholds (MLR < 0.665 and dMLR < 0.945) have better PFS and LRC. Of note, the current study is—to the best of our knowledge—the first to use LRC as a clinical endpoint.

With 183 patients the current cohort is slightly larger than the only other multicentre RWD in the field of blood biomarker research [13]. As mentioned above, Chen Jia et al. in their study used an LMR cutoff of 1.8, which—reversed—equals 0.556 [14]. This value is close to the one we found in our analysis (0.665). Similarly, a comprehensive review by Chan presented an LMR threshold range of 2.5–5.09 [16], the lower limit of which is—reversed—also in the same order of magnitude as our threshold. However, in this very same review, the upper range of values deviates substantially from our results. This discrepancy may have several reasons. First, our multicentre cohort of 183 patients exclusively comprises NSCLC UICC stages III, whereas other studies also include early-stage patients, such as UICC I-II [16,17,20]. This is important since inflammatory markers increase with higher tumour stage [31]. In fact, studies focusing exclusively on stage IV patients report cutoffs closer to ours, ranging between 0.25 [18] and 0.556 [14]. In our cohort, approximately 65% of patients are classified as UICC stages IIIb and IIIc, which have a prognosis similar to stage IV. Partially in line with this notion, the study by Chan—both on their own data and the meta-analysis—was able to define significant cutoffs only in the advanced stage subgroup, i.e., IIIb and IV [16]. However, the endpoint in this report was OS [16]. In this context, it seems important to be aware of the fact that the chosen clinical endpoint may also influence the cutoff value. While the reports published thus far chose PFS and/or OS [13,14,16,17], ALLSTAR adds another layer of complexity by also including LRC as an endpoint in the context of the MLR. Two other studies—one in surgical [32] and the other in chemoradiotherapy [33] patients—using LRC as an endpoint investigate the NLR as a biomarker to guide follow-up procedures. Apart from differences in the patient population and the mode of therapy, the type of blood biomarker used as a predictor, i.e., NLR, also hampers further comparisons to our study. A second reason for discrepancies in threshold values could be differences in the histology and molecular status of the tumours. While, for example, Chan et al. in their study included only EGFR-positive patients [16], the ALLSTAR cohort consists of less than 5% EGFR patients, since these patients are known for their limited benefit from immunotherapy maintenance after chemoradiotherapy. Thirdly, the applied therapeutic modes may also modify the threshold as these agents have multifaceted impacts on the tumours and their microenvironments. In the current analysis, all patients received curatively intended radiation [26,30] compared to other studies with 20% [14] to 30% [16]. In this respect, the widespread application of immunotherapy in daily clinical routine marks a watershed. While in the study by Chen, for example, with the threshold closest to ours, immunotherapy was used for all patients [14], other data were generated in the pre-immunotherapy era and are therefore less comparable to ours [16,18,20]. Given the above-mentioned reasons, we not only agree with Chen et al., who already pointed out that blood biomarker thresholds should be applied depending on UICC stage and co-morbidities [14], but would like to extend the panel of criteria to pathological findings, therapeutic strategies and the chosen clinical endpoints. As for OS, we were unable to detect a cutoff value for any of the analysed blood biomarkers that was stringently correlated with this endpoint, both in the whole cohort and the Durvalumab subgroup (see Appendix A).

The current study is the first to report dMLR as a biomarker for clinical outcomes. In coherence with the results on the MLR, this biomarker also predicted better LRC and PFS for those patients below the threshold, which strengthens the MLR results. The only other report presenting the dMLR as a blood biomarker demonstrated its predictive potential with respect to pulmonary toxicity [13], meaning that direct comparisons with the current analysis are not possible.

What some of the above-mentioned studies [14,16,17,18,20] have in common with ALLSTAR is the fact that patients with MLR and dMLR values below the threshold at baseline fare better in terms of clinical outcome. On a mechanistic level this finding can potentially be explained by the physiological function of lymphocytes and macrophages in the tumour microenvironment. An increase in lymphocyte number has been correlated with immune response to micrometastatic tumour invasion [23]. Moreover, radiotherapy promotes CD8+-T-lymphocyte mediated tumour cell killing via ICAM-1 and MIC A/B so that inflammatory remodelling of the microenvironment leads to increased tumour cell death (reviewed by Donlon [34]). The radioimmunobiological notion of an elevated antitumorigenic lymphocyte load is corroborated by our results. Patients with an MLR and dMLR below the threshold, which equals a high lymphocyte number, show better clinical outcomes (Figure 2 and Figure 3). As for the second cell type involved, i.e., monocytes, the literature reports that an increase deteriorates the prognosis [35]. A feature of tumour progression seems to be enhanced recruitment of this cell type, which are precursors of tumour-associated macrophages (TAMs) [22,36]. Hence—in line with our findings—a high MLR might be a surrogate marker for an immunosuppressive tumour microenvironment.

An inherent limit of biomarker studies like this is the fact that, at present, there are no generally accepted standard cutoffs that could be readily applied to different patient populations. Although the MLR cutoff used in the present study is very close to published data [14], an external validation of our findings is warranted. Moreover, ALLSTAR is a multicentric RWD study, which entails some heterogeneity in the patient population. However, since ALLSTAR includes inoperable UICC stage III patients only, the degree of variance in clinical features is lower than in most other studies [14,16,17,18,20].

## 5. Conclusions

The current study on patients from the multicentre ALLSTAR registry demonstrated the predictive potential of the MLR and dMLR for LRC and PFS. These blood biomarkers can be easily retrieved from pre-treatment blood samples and therefore readily integrated into clinical routines, which is important in the context of a rising awareness of health costs. Although a validation in larger series is warranted, the MLR and dMLR could be part of a panel of biomarkers representing systemic inflammation in NSCLC.

## Figures and Tables

**Figure 1 jcm-14-08876-f001:**
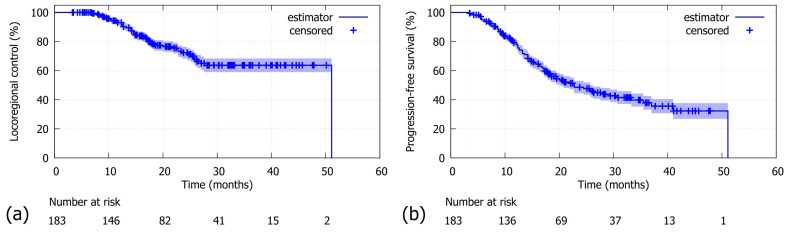
Whole cohort (N = 183): (**a**) For locoregional control (LRC), the 2-year LRC was 72.3% (median: 51.1 months). (**b**) For progression-free survival (PFS), the 2-year PFS was 48.6% (median: 22.7 months).

**Figure 2 jcm-14-08876-f002:**
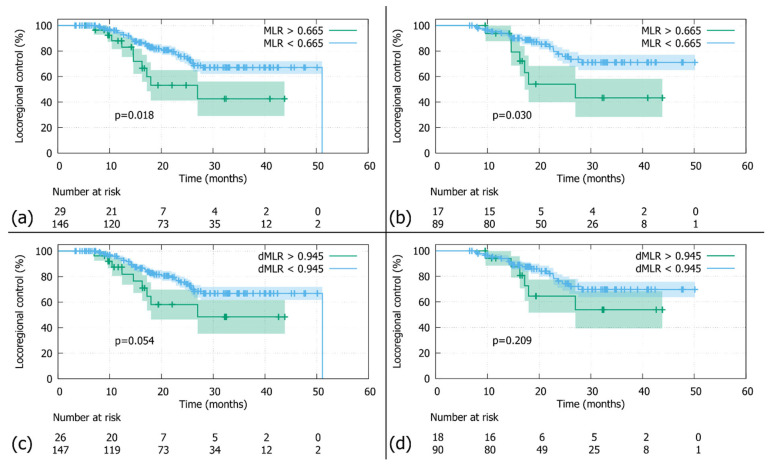
Locoregional control (LRC): (**a**) In the whole cohort stratified by the MLR threshold (0.665), patients with MLRs above the cutoff had significantly worse LRCs than those below (N = 175; log-rank *p*-value 0.018). (**b**) In the Durvalumab subgroup stratified by the MLR threshold (0.665), patients with MLRs above the cutoff had significantly worse LRCs than those below (N = 106; log-rank *p*-value 0.030). (**c**) In the whole cohort stratified by the dMLR threshold (0.945), patients with dMLRs above the cutoff showed a tendency towards inferior LRCs, which did not reach statistical significance (N = 173; log-rank *p*-value 0.054). (**d**) In the Durvalumab subgroup stratified by the dMLR threshold (0.945), patients with dMLRs above the cutoff had nominally inferior LRCs, which did not reach statistical significance (N = 108; log-rank *p*-value = 0.209).

**Figure 3 jcm-14-08876-f003:**
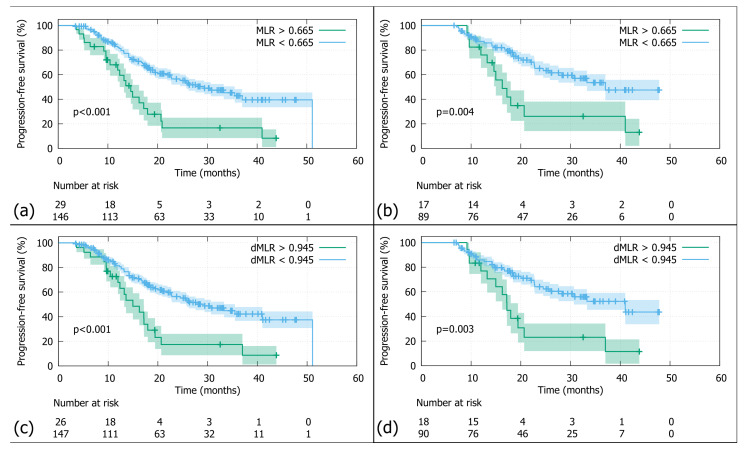
Progression-free survival (PFS): (**a**) In the whole cohort stratified by the MLR threshold (0.665), patients with MLRs above the cutoff had significantly worse PFS than those below (N = 175; log-rank *p*-value < 0.001). (**b**) In the Durvalumab subgroup stratified by the MLR threshold (0.665), patients with MLRs above the cutoff had significantly worse PFS than those below (N = 106; log-rank *p*-value < 0.001). (**c**) In the whole cohort stratified by the dMLR threshold (0.945), patients with dMLRs above the cutoff had significantly worse PFS than those below (N = 173; log-rank *p*-value < 0.001). (**d**) In the Durvalumab subgroup stratified by the dMLR threshold (0.945), patients with dMLRs above the cutoff had significantly worse PFS than those below (N = 108; log-rank *p*-value = 0.003).

**Table 1 jcm-14-08876-t001:** Baseline characteristics of the whole cohort (N = 183) and the Durvalumab subgroup (N = 112).

Patients				
		All	Durvalumab	*p*-Value
		N = 183 (%)	N = 112 (%)	
Sex	Male	111 (60.7)	70 (62.5)	0.990
	Female	72 (39.3)	42 (37.5)	
Age	Median	67.3	67.4	0.899
	Range	36.4–90.7	40.6–83.9	
Smoking	Never	13 (7.1)	10 (8.9)	0.703
	Ex	104 (56.8)	63 (56.3)	
	Current	66 (36.1)	39 (34.8)	
ECOG	0–1	171(93.4)	106 (94.6)	0.375
	2–3	12 (6.6)	6 (5.4)	
Histology	AC	95 (51.9)	61 (54.5)	0.559
	SCC	77 (42.1)	47 (42.0)	
	NOS	11 (6.0)	4 (3.5)	
PD-L1	<1%	40 (21.9)	12 (10.7)	0.097
	>1%	128 (69.9)	93 (83.0)	
	Unknown	15 (8.2)	7 (6.3)	
T-stage	Tis	2 (1.1)	1 (0.9)	0.475
	1	26 (14.2)	18 (16.1)	
	2	21 (11.5)	13 (11.6)	
	3	57 (31.1)	38 (33.9)	
	4	76 (41.6)	41 (36.6)	
	Unknown	1 (0.5)	1 (0.9)	
N-stage	0	14 (7.7)	8 (7.1)	0.883
	1	16 (8.7)	7 (6.3)	
	2	112 (61.2)	76 (67.9)	
	3	41 (22.4)	21 (18.7)	
M-stage	0	183 (100.0)	112 (100.0)	1.000
	1	0 (0.0)	0.0 (0.0)	
UICC	IIIa	64 (34.9)	41 (36.6)	0.631
	IIIb	83 (45.4)	52 (46.4)	
	IIIc	36 (19.7)	19 (17.0)	

AC = adenocarcinoma, SCC = squamous cell carcinoma, NOS = not otherwise specified, PD-L1 = Programmed Cell Death 1 Ligand 1.

**Table 2 jcm-14-08876-t002:** The treatment characteristics of the whole cohort (N = 183) and the Durvalumab subgroup (N = 112).

Treatment				
		All N = 183 (%)	Durvalumab N = 112 (%)	*p*-Value
Treatment sequence	cCRT	46 (25.1)	30 (26.8)	0.384
	sCRT	126 (68.9)	81 (72.3)	
	RIT	11 (6.0)	1 (0.9)	
Immune Checkpoint Inhibitors	Durvalumab	112 (61.2)	112 (100.0)	<0.001
	Other ICI *	16 (8.7)	0 (0.0)	
	No ICI	55 (30.1)	0 (0.0)	
RT-Technique	VMAT/IMRT	178 (97.3)	109 (97.3)	0.659
	3D	5 (2.7)	3 (2.7)	
Tumour	EQD2 median	65.0	67.1	0.051
	EQD2 range	0.0–100.0	0.0–100.0	
	GTV median	48.4	44.4	0.242
	GTV range	0.0–784.1	0.0–483.8	
Lymph nodes	EQD2 median	57.3	57.3	0.309
	EQD2 range	0.0–81.3	0.0–70.0	
	GTV median	29.0	26.0	0.364
	GTV range	0.0–473.0	0.0–293.3	
Elective nodal irradiation	EQD2 median	32.5	48.8	0.832
	EQD2 range	32.5–60.0	32.5–60.0	
	GTV median	90.0	79.0	0.523
	GTV range	0.0–429.0	0.0–429.0	

* Other ICI: Nivolumab, Pembrolizumab, Atezolizumab; cCRT = concomitant chemoradiotherapy; sCRT = sequential chemoradiotherapy; RIT = radioimmunotherapy; EQD2 = biologically equivalent dose in 2 Gy fractions; GTV = gross tumour volume; IMRT = intensity modulated radiotherapy; VMAT = volumetric arc therapy.

**Table 3 jcm-14-08876-t003:** (**a**) Locoregional control (LRC): Multivariate analysis for MLR and clinical characteristics revealed histology as the only significant parameter that impacted LRC in the whole cohort (*p*-value = 0.004; corrected *p*-value = 0.02; HR 2.6; 95% CI 1.35–4.89) as well as in the Durvalumab subgroup (*p*-value < 0.01; corrected *p*-value 0.047; HR 3.2; 95% CI 1.3–7.7). (**b**) Locoregional control (LRC): Multivariate analysis for MLR and treatment characteristics revealed MLR as the only significant parameter that impacted LRC in the whole cohort (*p*-value = 0.012; corrected *p*-value = 0.070; HR 0.38; 95% CI 0.18–0.81) and the Durvalumab subgroup (*p*-value < 0.05; corrected *p*-value = 0.15; HR 0.4; 95% CI 0.16–0.998), in which GTV_Lymphnodes_ was also significantly associated with better LRC (*p*-value = 0.01; corrected *p*-value = 0.07; HR = 1.0095; 95% CI 1.0022–1.017). (**c**) Locoregional control (LRC): Multivariate analysis for dMLR and clinical characteristics revealed histology as the only significant parameter that impacted LRC in the whole cohort (*p*-value < 0.003; corrected *p*-value = 0.013; HR 2.75; 95% CI 1.4–5.3) and the Durvalumab subgroup (*p*-value = 0.007; corrected *p*-value = 0.03; HR 3.3; 95% CI 1.4–7.9). (**d**) Locoregional control (LRC): Multivariate analysis for dMLR and treatment characteristics revealed dMLR as the only significant parameter that impacted LRC in the whole cohort (*p*-value = 0.019; corrected *p*-value = 0.11; HR = 0.396; 95% CI 0.18–0.86).

(**a**)								
**Baseline Characteristics**								
**All (N = 174; Events = 43)**								
**Factor (# of Cases)**	**Univariate Analysis**				**Multivariate Analysis**			
	***p* Value**	**HR**	**95% CI**		***p* Value/BH adj.**	**HR**	**95% CI**	
			**Lower**	**Upper**			**Lower**	**Upper**
Age (min: 36.39; median: 67.58; max: 90.73)	0.388	0.987	0.959	1.016	0.37/0.37	0.986	0.96	1.02
Sex (female: 72; male: 102)	0.623	1.141	0.675	1.928	0.23/0.37	1.6	0.76	3.18
Histology (nonSCC: 101; SCC: 73)	<0.001	2.164	1.372	3.414	0.004/0.02	2.6	1.35	4.89
UICC (IIIa: 60; IIIbc: 114)	0.534	1.135	0.761	1.694	0.29/0.37	1.4	0.73	2.85
MLR (low: 145; high: 29)	0.044	2.084	1.021	4.253	0.11/0.28	1.8	0.8	3.7
**Durvalumab (N = 105; Events = 25)**								
**Factor (# of Cases)**	**Univariate Analysis**				**Multivariate Analysis**			
	***p* Value**	**HR**	**95% CI**		***p* Value/BH adj.**	**HR**	**95% CI**	
			**Lower**	**Upper**			**Lower**	**Upper**
Age (min: 36.9; median: 67.7; max: 84.13)	0.486	0.984	0.944	1.0266	0.74/0.74	1.4	0.95	1.04
Sex (female: 47; male: 58)	0.449	0.724	0.315	1.667	0.52/0.65	0.99	0.54	3.4
Histology (nonSCC: 61; SCC: 44)	<0.001	3.808	1.891	7.667	<0.01/0.047	3.2	1.3	7.7
UICC (IIIa: 37; IIIbc: 68)	0.116	1.646	0.883	3.070	0.29/0.49	1.6	0.65	4
MLR (low: 88; high: 17)	0.033	2.654	1.079	6.529	0.12/0.29	2	0.8	5
(**b**)								
**Treatment Characteristics**								
**All (N = 162; Events = 42)**								
**Factor(# of Cases)**	**Univariate Analysis**				**Multivariate Analysis**			
	***p* Value**	**HR**	**95% CI**		***p* Value/BH adj.**	**HR**	**95% CI**	
			**Lower**	**Upper**			**Lower**	**Upper**
GTV Tumour (min: 0.24; median: 48.4; mean: 87.08; max: 589.3)	0.1	1.002	0.999	1.004	0.57/0.69	1.0008	0.998	1.0037
GTV Lymphnodes (min: 0; median: 29.2; mean: 56.65; max: 473)	0.06	1.004	0.9999	1.008	0.15/0.25	1.0035	0.9987	1.0084
EQD2 Tumour (min: 24.8; median: 63.81; mean: 63.81; max: 100.0)	0.06	0.977	0.95	1.001	0.17/0.25	0.9744	0.939	1.01
EQD2 Lymphnodes (min: 0; median: 57.29; mean: 48.92; max: 70.0)	0.3	1.01	0.991	1.029	0.12/0.25	1.0195	0.9947	1.045
cCRT vs. sCRT (cCRT: 38; sCRT: 124)	0.5	0.78	0.38	1.6	0.84/0.84	1.08	0.51	2.28
MLR (high: 26; low: 136)	0.01	0.42	0.21	0.86	0.012/0.07	0.38	0.18	0.81
**Durvalumab (N = 103; Events = 25)**								
**Factor (# of Cases)**	**Univariate Analysis**				**Multivariate Analysis**			
	***p* Value**	**HR**	**95% CI**		***p* Value/** **BH adj.**	**HR**	**95% CI**	
			**Lower**	**Upper**			**Lower**	**Upper**
GTV Tumour (min: 0.24; median: 46; mean: 69.68; max: 483)	0.7	1.001	0.997	1.005	0.62/0.62	1.0012	0.997	1.006
GTV Lymphnodes (min: 0; median: 26.7; mean: 45.89; max: 285)	0.007	1.008	1.002	1.014	0.01/0.07	1.0095	1.0022	1.017
EQD2 Tumour (min: 32.5; median: 66; mean: 66.51; max: 100)	0.3	0.978	0.94	1.022	0.4/0.5	0.977	0.923	1.03
EQD2 Lymphnodes (min: 0; median: 57.29; mean: 50.09; max: 70))	0.3	1.014	0.983	1.046	0.29/0.44	1.019	0.98	1.05
cCRT vs. sCRT (cCRT: 23; sCRT: 80)	0.9	1.033	0.387	2.755	0.2/0.4	2.032	0.685	6.03
MLR (high: 18; low: 85)	0.08	0.46	0.19	1.113	<0.05/0.15	0.4	0.16	0.9983
(**c**)								
**Baseline Characteristics**								
**All (N = 172; Events = 42)**								
**Factor (# of Cases)**	**Univariate Analysis**				**Multivariate Analysis**			
	***p* Value**	**HR**	**95% CI**		***p* Value/** **BH adj.**	**HR**	**95% CI**	
			**Lower**	**Upper**			**Lower**	**Upper**
Age (min: 36.39; median: 67.36; max: 90.73)	0.5	0.99	0.96	1.02	0.51/0.51	0.99	0.96	1.02
Sex (female: 69; male: 103)	0.04	2.1	1.036	4.3	0.18/0.41	1.66	0.8	3.5
Histology (nonSCC: 100; SCC: 72)	<0.001	3.1	1.66	5.9	<0.003/0.013	2.75	1.4	5.3
UICC (IIIa: 62; IIIbc: 110)	0.3	1.47	0.75	2.9	0.3/0.4	1.4	0.7	2.8
dMLR (high: 26; low: 146)	0.06	0.49	0.23	1.03	0.3/0.4	0.68	0.32	1.5
**Durvalumab (N = 107; events = 25)**								
**Factor (# of Cases)**	**Univariate Analysis**				**Multivariate Analysis**			
	***p* Value**	**HR**	**95% CI**		***p* Value/** **BH adj.**	**HR**	**95% CI**	
			**Lower**	**Upper**			**Lower**	**Upper**
Age (min: 36.39; median: 67.19; max: 84.13)	0.44	1.004	0.96	1.045	0.86/0.86	0.996	0.96	1.03
Sex (female: 46; male: 61)	0.65	1.8	0.75	4.3	0.49/0.72	1.37	0.56	3.39
Histology (nonSCC: 63; SCC: 44)	0.001	3.6	1.6	8.4	<0.007/0.03	3.3	1.4	7.9
UICC (IIIa: 39; IIIbc: 68)	0.2	1.7	0.7	4.1	0.28/0.7	1.6	0.66	4.1
dMLR (high: 18; low: 89)	0.2	0.56	0.22	1.4	0.58/0.72	0.76	0.3	1.96
(**d**)								
**Treatment Characteristics**								
**All (N = 160, Events = 41)**								
**Factor (# of Cases)**	**Univariate Analysis**				**Multivariate Analysis**			
	***p* Value**	**HR**	**95% CI**		***p* Value/** **BH adj.**	**HR**	**95% CI**	
			**Lower**	**Upper**			**Lower**	**Upper**
GTV Tumour (min: 0.24; median: 48; mean: 86.08; max: 589.3)	0.1	1.002	0.999	1.004	0.6/0.73	1.0008	0.9978	1.0037
GTV Lymphnodes (min: 0; median: 27.65; mean: 49.77; max: 303.6)	0.01	1.006	1.001	1.011	0.06/0.18	1.0055	0.9998	1.01
EQD2 Tumour (min: 24.8; median: 65; mean: 64.2; max: 100)	0.05	0.976	0.95	0.9996	0.21/0.31	0.977	0.94	1.013
EQD2 Lymphnodes (min: 0; median: 57.29; mean: 49.2; max: 70)	0.4	1.008	0.9899	1.027	0.16/0.31	1.018	0.99	1.043
cCRT vs. sCRT (cCRT: 39; sCRT: 121)	0.5	0.79	0.386	1.62	0.83/0.83	1.08	0.515	2.285
dMLR (high: 23; low: 137)	0.03	0.455	0.213	0.957	0.019/0.11	0.396	0.18	0.86
**Durvalumab (N = 105; events = 25)**								
**Factor (# of Cases)**	**Univariate Analysis**				**Multivariate Analysis**			
	***p* Value**	**HR**	**95% CI**		***p* Value/** **BH adj.**	**HR**	**95% CI**	
			**Lower**	**Upper**			**Lower**	**Upper**
GTV Tumour (min: 0.24; median: 45; mean: 67; max: 483.8)	0.6	1.001	0.997	1.005	0.59/0.59	1.0013	0.997	1.006
GTV Lymphnodes (min: 0; median: 24.6; mean: 45.04; max: 285)	0.006	1.008	1.002	1.014	0.01/0.065	1.0094	1.0021	1.017
EQD2 Tumour (min: 32.5; median: 67.1; mean: 66.61; max: 100)	0.3	0.976	0.93	1.02	0.38/0.46	0.975	0.92	1.03
EQD2 Lymphnodes (min: 0; median: 57.29; mean: 50.50; max: 70)	0.3	1.013	0.986	1.041	0.27/0.4	1.02	0.99	1.056
cCRT vs. sCRT (sCRT: 25; sCRT: 80)	0.9	1.06	0.3976	2.83	0.2/0.4	2.01	0.683	5.93
dMLR (high: 17; low: 88)	0.2	0.568	0.227	1.43	0.16/0.4	0.499	0.189	1.32

(**a**) MLR = monocyte–lymphocyte ratio, BH = Benjamini–Hochberg. (**b**) EQD2 = biologically equivalent dose in 2 Gy fractions, MLR = monocyte–lymphocyte ratio, GTV = gross tumour volume, cCRT = concomitant chemoradiotherapy, sCRT = sequential chemoradiotherapy, BH = Benjamini–Hochberg. (**c**) dMLR = monocyte–lymphocyte ratio, BH = Benjamini–Hochberg. (**d**) EQD2 = biologically equivalent dose in 2 Gy fractions, dMLR = derived monocyte–lymphocyte ratio, GTV = gross tumour volume, cCRT = concomitant chemoradiotherapy, sCRT sequential chemoradiotherapy, BH = Benjamini–Hochberg.

**Table 4 jcm-14-08876-t004:** (**a**) Progression-free survival (PFS): In the whole cohort, multivariate analysis for MLR and clinical characteristics showed that UICC (*p*-value = 0.004; corrected *p*-value < 0.01; HR 2.1; 95% CI 1.28–3.56) and MLR (*p*-value < 0.001; corrected *p*-value = 0.004; HR 0.43; 95% CI 0.26–0.7) significantly impacted PFS. This finding was corroborated in the Durvalumab subgroup with UICC (*p*-value = 0.04; corrected *p*-value = 0.073; HR 1.91; 95% CI 1.02–3.6) and MLR (*p*-value = 0.014; corrected *p*-value = 0.073; HR 0.4; 95% CI 0.22–0.85) as significant parameters. Additionally, histology was also significant (*p*-value = 0.044; corrected *p*-value = 0.073; HR 1.91; 95% CI 1.03–4.3). (**b**) Progression-free survival (PFS): Multivariate analysis for MLR and treatment characteristics revealed EQD2_Tumour_ (*p*-value < 0.03; corrected *p*-value = 0.08; HR 0.97; 95% CI 0.95–0.9973) and MLR (*p*-value < 0.001; corrected *p*-value < 0.005; HR 0.39; 95% CI 0.22–0.68) to have a significant impact on PFS in the whole cohort. In the Durvalumab subgroup, MLR was the only factor to remain significant (*p*-value = 0.0059; corrected *p*-value 0.035; HR 0.365; 95% CI 0.178–0.75). (**c**) Progression-free survival (PFS): Multivariate analysis for dMLR and clinical characteristics revealed that UICC (*p*-value < 0.009; corrected *p*-value = 0.023; HR 1.96; 95% CI 1.18–3.2) and dMLR (*p*-value = 0.04; corrected *p*-value = 0.02; HR 0.45; 95% CI 0.26–0.78) were significant factors impacting PFS in the whole cohort. This result was corroborated in the Durvalumab subgroup: UICC (*p*-value = 0.048; corrected *p*-value 0.12; HR 2.01; 95% CI 1.006–4.0) and dMLR (*p*-value = 0.018; corrected *p*-value = 0.088; HR 0.4; 95% CI 0.22–0.86). (**d**) Progression-free survival (PFS): Multivariate analysis for dMLR and clinical characteristics revealed that dMLR was the only significant factor to impact PFS in the whole cohort (*p*-value = 0.002; corrected *p*-value = 0.014; HR 0.41; 95% CI: 0.2296–0.7264) as well as in the Durvalumab subgroup (*p*-value = 0.0017; corrected *p*-value = 0.01; HR 0.314; 95% CI 0.15–0.65).

(**a**)								
** Baseline Characteristics **								
** All (N = 174; Events = 85) **								
** Factor (# of Cases) **	** Univariate Analysis **				** Multivariate Analysis **			
	** * p * ** ** Value **	** HR **	** 95% CI **		** * p * ** ** Value/BH** **adj. **	** HR **	** 95% CI **	
			** Lower **	** Upper **			** Lower **	** Upper **
Age (min: 36.39; median: 67.58; max: 90.73)	0.3	0.99	0.959	1.016	0.12/0.17	0.98	0.96	1.005
Sex (female: 72; male: 102)	1	0.99	0.65	1.5	0.48/0.48	0.85	0.54	1.34
Histology (nonSCC: 101; SCC: 73)	0.1	45,748	0.9	2.1	0.14/0.17	1.4	0.9	2.2
UICC (IIIa: 60; IIIbc: 114)	<0.01	1.9	1.17	3.2	0.004/<0.01	2.1	1.28	3.56
MLR (high: 29; low: 145)	<0.001	0.42	0.25	0.69	<0.001/0.004	0.43	0.26	0.7
** Durvalumab (N = 105; Events = 44) **								
** Factor (# of Cases) **	** Univariate Analysis **				** Multivariate Analysis **			
	** * p * ** ** Value **	** HR **	** 95% CI **		** * p * ** ** Value/BH adj. **	** HR **	** 95% CI **	
			** Lower **	** Upper **			** Lower **	** Upper **
Age (min: 36.39; median 67.7; max: 84.13)	0.8	1.005	0.97	1.037	0.59/0.59	0.991	0.96	1.02
Sex (female: 47; male: 58)	0.9	1.05	0.58	1.91	0.59/0.59	0.84	0.44	1.6
Histology (nonSCC: 61; SCC: 44)	0.04	1.8	1.02	3.34	0.044/0.073	1.91	1.02	3.6
UICC (IIIa: 37; IIIbc: 68)	0.06	1.905	0.97	3.7	0.04/0.074	2.1	1.03	4.3
MLR (high: 17; low: 88)	0.01	0.42	0.22	0.83	0.014/0.073	0.4	0.22	0.85
(**b**)								
** Treatment Characteristics **								
** All (N = 162, Events = 80) **								
** Factor (# of Cases) **	** Univariate Analysis **				** Multivariate Analysis **			
	** * p * ** ** Value **	** HR **	** 95% CI **		** * p * ** ** Value/** **BH adj. **	** HR **	** 95% CI **	
			** Lower **	** Upper **			** Lower **	** Upper **
GTV Tumour (min: 0.24; median: 48.41; mean: 87.08; max: 589.3)	0.1	1.001	0.9997	1.003	0.96/0.96	0.9999	0.9978	1.002
GTV Lymphnodes (min: 0; median: 29.2; mean: 56.65; max: 473)	0.007	1.004	1.001	1.007	0.04/0.08	1.0035	1.0001	1.0068
EQD2 Tumour (min: 24.8; median: 63.81; mean: 63.81; max: 100)	0.003	0.975	0.959	0.9915	<0.03/0.08	0.97	0.95	0.9973
EQD2 Lymphnodes (min: 0; median: 57.29; mean: 48.92; max: 70)	0.6	1.003	0.9913	1.015	0.2/0.32	1.0092	0.99	1.024
cCRT vs. sCRT (cCRT: 38; sCRT: 124)	0.3	0.75	0.45	1.253	0.9/0.96	0.97	0.57	1.65
MLR (high: 26: low: 136)	0.003	0.46	0.268	0.774	<0.001/<0.005	0.39	0.22	0.68
** Durvalumab (N = 103; Events = 43) **								
** Factor (# of Cases) **	** Univariate Analysis **				** Multivariate Analysis **			
	** * p * ** ** Value **	** HR **	** 95% CI **		** * p * ** ** Value/** **BH adj. **	** HR **	** 95% CI **	
			** Lower **	** Upper **			** Lower **	** Upper **
GTV Tumour (min: 0.24; median: 46; mean: 69.68; max: 483.8)	0.3	1.001	0.998	1.004	0.57/0.69	1.001	0.9975	1.0045
GTV Lymphnodes (min: 0; median: 26.7; mean: 45.89; max: 285)	0.4	1.003	0.9969	1.008	0.1/0.2	1.005	0.9989	1.012
EQD2 Tumour (min: 32.5; median: 66; mean: 66.51; max: 100)	0.2	0.98	0.945	1.011	0.23/0.35	0.9775	0.94	1.015
EQD2 Lymphnodes (min: 0; median: 57.29; mean: 50.09; max: 70)	0.7	0.997	0.98	1.012	0.75/0.75	1.0029	0.985	1.021
cCRT vs. sCRT (cCRT: 23; sCRT: 80)	0.4	1.378	0.6	3.070	0.089/0.21	2.2	0.886	5.46
MLR (high: 18: low: 85)	0.01	0.44	0.22	0.857	0.0059/0.035	0.365	0.178	0.75
(**c**)								
** Baseline Characteristics **								
** All (N = 172; Events = 83) **								
** Factor (# of Cases) **	** Univariate Analysis **				** Multivariate Analysis **			
	** * p * ** ** Value **	** HR **	** 95% CI **		** * p * ** ** Value/BH** **adj. **	** HR **	** 95% CI **	
			** Lower **	** Upper **			** Lower **	** Upper **
Age (min: 36.39; median: 67.36; max: 90.73)	0.4	0.9905	0.97	1.013	0.13/0.22	0.98	0.96	1.005
Sex (female: 69; male: 103)	0.21	1.001	0.64	1.56	0.53/0.53	0.86	0.55	1.36
Histology (nonSCC: 100; SCC: 72)	0.1	1.4	0.9	2.16	0.31/0.39	1.27	0.8	2.02
UICC (IIIa: 62; IIIbc: 110)	0.02	1.82	1.1	2.97	<0.009/0.023	1.96	1.18	3.2
dMLR (high: 26; low: 146)	<0.001	0.42	0.25	0.7	0.04/0.02	0.45	0.26	0.78
** Durvalumab (N = 107; Events = 46) **								
** Factor (# of Cases) **	** Univariate Analysis **				** Multivariate Analysis **			
	** * p * ** ** Value **	** HR **	** 95% CI **		** * p * ** ** Value/BH adj. **	** HR **	** 95% CI **	
			** Lower **	** Upper **			** Lower **	** Upper **
Age (min: 36.39; median: 67.19; max: 84.13)	0.9	1.002	0.97	1.03	0.41/0.51	0.986	0.95	1.019
Sex (female: 46; male: 61)	0.8	1.07	0.6	1.9	0.77/0.77	0.91	0.49	1.69
Histology (nonSCC: 63; SCC: 44)	0.08	1.67	0.93	2.97	0.23/0.39	1.46	0.79	2.7
UICC (IIIa: 39; IIIbc: 68)	0.06	1.86	0.97	3.55	0.048/0.12	2.01	1.006	4
dMLR (high: 18: low: 89)	0.004	0.39	0.21	0.75	0.018/0.088	0.4	0.22	0.86
(**d**)								
** Treatment Characteristics **								
** All (N = 160; Events = 78) **								
** Factor (# of Cases) **	** Univariate Analysis **				** Multivariate Analysis **			
	** * p * ** ** Value **	** HR **	** 95% CI **		** * p * ** ** Value/** **BH adj. **	** HR **	** 95% CI **	
			** Lower **	** Upper **			** Lower **	** Upper **
GTV Tumour (min: 0.24; median: 48; mean: 86.08; max: 589.3)	0.1	1.001	0.9997	1.003	0.9/0.9	0.9999	0.998	1.002
GTV Lymphnodes (min: 0; median: 27.65; mean: 49.77; max: 303.6)	0.3	1.002	0.998	1.006	0.7/0.84	1.0009	0.9964	1.0053
EQD2 Tumour (min: 24.8; median: 65; mean: 64.2; max: 100)	0.01	0.978	0.96	0.995	0.19/0.058	0.97	0.947	0.995
EQD2 Lymphnodes (min: 0; median: 57.29; mean: 49.2; max: 70)	0.4	1.005	0.993	1.017	0.096/0.19	1.0125	0.9978	1.027
cCRT vs. sCRT (cCRT: 39; sCRT: 121)	0.1	0.68	0.41	1.123	0.54/0.81	0.847	0.4977	1.441
dMLR (high: 23; low: 137)	0.003	0.44	0.257	0.762	0.002/0.014	0.41	0.2296	0.7264
** Durvalumab (N = 105; events = 45) **								
** Factor (# of Cases) **	** Univariate Analysis **				** Multivariate Analysis **			
	** * p * ** ** Value **	** HR **	** 95% CI **		** * p * ** ** Value/BH adj. **	** HR **	** 95% CI **	
			** Lower **	** Upper **			** Lower **	** Upper **
GTV Tumour (min: 0.24; median: 45; mean: 67.5; max: 483.8)	0.4	1.001	0.998	1.004	0.97/0.97	1.0001	0.9967	1.0035
GTV Lymphnodes (min: 0; median: 24.6; mean: 45.04; max: 285)	0.5	1.002	0.996	1.008	0.25/0.45	1.0037	0.997	1.01
EQD2 Tumour (min: 32.5; median: 67.1; mean: 66.61; max: 100)	0.2	0.977	0.946	1.01	0.21/0.45	0.976	0.9395	1.014
EQD2 Lymphnodes (min: 0; median: 57.29; mean: 50.5; max: 70)	0.9	0.9989	0.984	1.014	0.39/0.48	1.0078	0.9899	1.0259
cCRT vs. sCRT (cCRT: 25: sCRT: 80)	0.9	1.032	0.496	2.146	0.3/0.48	1.5434	0.68	3.5
dMLR (high: 17: low: 88)	0.003	0.3818	0.199	0.7322	0.0017/0.01	0.314	0.15	0.65

(**a**) MLR = monocyte–lymphocyte ratio, BH = Benjamini–Hochberg. (**b**) MLR = monocyte–lymphocyte ratio, GTV = gross tumour volume, cCRT = concomitant chemoradiotherapy, sCRT sequential chemoradiotherapy, BH = Benjamini–Hochberg. (**c**) dMLR = monocyte–lymphocyte ratio, BH = Benjamini–Hochberg. (**d**) MLR = monocyte–lymphocyte ratio, BH = Benjamini–Hochberg. dMLR = derived monocyte–lymphocyte ratio, GTV = gross tumour volume, cCRT = concomitant chemoradiotherapy, sCRT = sequential chemoradiotherapy.

## Data Availability

The original contributions presented in this study are included in the article and the Appendix A. Further inquiries can be directed to the corresponding author.

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
