# Peer review of "MLR and dMLR Predict Locoregional Control and Progression-Free Survival in Unresectable NSCLC Stage III Patients: Results from the Austrian Radio-Oncological Lung Cancer Study Association Registry (ALLSTAR)"

_jcm, 2025, doi:10.3390/jcm14248876_

Round 1
Reviewer 1 Report
Comments and Suggestions for Authors
This study based on the multicenter ALLSTAR real-world cohort, investigates the predictive value of MLR and dMLR for locoregional control and progression-free survival in unresectable stage III NSCLC. The research topic is clinically meaningful and aligns with the current interest in inflammatory markers as prognostic biomarkers. The findings demonstrate certain potential for clinical application. However, several minor issues remain to be addressed:
The authors’ affiliations appear incomplete. Please provide full details, including institution, city, and country.
The current abstract is too long and seems to exceed 400 words. Please shorten it.
In Table 1, it is recommended to add the group names in the table header. In addition, since no patients had unknown N-stage, this category could be removed.
In Table 1, it is unclear why patients with M-stage 1 were classified as stage III.
Based on the p-values in Figure 2c and 2d, the results do not seem statistically significant. The figure legends should avoid overly definitive statements.
The abbreviations “UVA” and “MVA” lack full spelling. Based on the tables, these appear to refer to “univariate analysis” and “multivariate analysis.” Please provide the full terms at their first mention.
The descriptions in Tables 3 and 4 do not fully correspond to those in the main text. It is recommended to present the results in the form of HR (95% CI) in the tables.
The analyses in the manuscript are limited to PFS and LRC. It is recommended to discuss the potential value regarding OS in the Discussion section.
Author Response
We would like to express our gratitude to reviewer 1 for the effort of providing a thoughtful commentary on our paper. Please find our point-by-point answers to the questions below (insertions in the manuscript text in blue font).
Reviewer 1
This study based on the multicenter ALLSTAR real-world cohort, investigates the predictive value of MLR and dMLR for locoregional control and progression-free survival in unresectable stage III NSCLC. The research topic is clinically meaningful and aligns with the current interest in inflammatory markers as prognostic biomarkers. The findings demonstrate certain potential for clinical application. However, several minor issues remain to be addressed:
Comment 1. The authors’ affiliations appear incomplete. Please provide full details, including institution, city, and country.
Response 1. Sorry for this inaccuracy. We corrected this error.
a These Authors contributed equally.
1 Paracelsus Medical University, Department of Radiation Oncology, 5020 Salzburg, Austria
2 Klinikum Ottakring, 1160 Vienna, Austria
3 Medical University Vienna, Department of Radiation Oncology, Comprehensive Cancer Centre, 1090 Vienna, Austria
4 Vienna Medical University Innsbruck, Department of Radiation Oncology, Comprehensive Cancer Centre, 6020 Innsbruck, Austria
5 Klinikum Hietzing-Rosenhügel, 1130 Vienna, Austria
6 Ordensklinikum, Department of Radiation Oncology, and Johannes Kepler University, 4020 Linz, Austria
7 Paracelsus Medical University, Department of Pulmonology, 5020 Salzburg, Austria
8 Institute of Research and Development of Advanced Radiation Technologies (radART), Paracelsus Medical University, 5020 Salzburg, Austria
* Correspondence: f.zehentmayr@salk.at; Tel.: +43 57255 58915.
Comment 2. The current abstract is too long and seems to exceed 400 words. Please shorten it.
Response 2. The initial abstract had 368 words. On re-reading it, however, we detected some redundant passages, which we shortened in accordance with the reviewer’s suggestion so that it has 273 words now.
Background: Despite improved clinical outcomes with the PACIFIC regimen, lung cancer is still one of the most deadly tumours. As demonstrated by the PACIFIC trial In this context biomarker driven patient selection is crucial. While treatment decisions based on programmed death ligand-1 (PD-L1) and mutational status have become clinical routine, tests for biomarkers available from pre-therapeutic blood samples are currently a topic of scientific interest. Methods: This analysis was conducted in patients from the ALLSTAR RWD study, which is a nationwide, prospective registry for inoperable non-small cell lung cancer (NSCLC) stage III. Patients were amenable if they had a full routine pre-treatment blood sample, from which the following biomarkers were extracted: neutrophil-to-lymphocyte ratio (NLR), derived neutrophil-to-lymphocyte ratio (dNLR), platelet-to-lymphocyte ratio (PLR), monocyte-to-lymphocyte ratio (MLR), derived monocyte-to-lymphocyte ratio (dMLR) and lactate dehydrogenase (LDH) levels. The intention was to find a cutoff for each of these biomarkers to predict loco-regional control (LRC), progression free survival (PFS) and overall survival (OS). Results: Full blood sample data were available from 183 patients, which comprised a subcohort of 112 Durvalumab patients. From the above mentioned biomarkers, MLR and dMLR demonstrated their predictive potential with cutoff values of 0.665 and 0.945, respectively. Stratifying the whole cohort by means of these cutoffs demonstrated significantly better loco-regional control for patients below threshold both in the whole cohort (N=175; 2-year LRC rates of 55.7% vs. 75.5%; log-rank p-value=0.018) and the Durvalumab subgroup (N=106; 2-year LRC rates of 57.5% vs. 77.3%; log-rank p-value = 0.030). Similar findings were observed for PFS in the whole cohort (N=175; 2-year PFS rates 20.5% vs. 56.1%; log-rank p-value p<0.001) and the Durvalumab subgroup (N=106; 2-year PFS of 31.2% vs. 64.6%, log-rank p-value <0.001). dMLR could also significantly predict PFS (N=173; 2-year PFS of 17.4% vs. 56.3%; log-rank p-value <0.001), which was corroborated in the Durvalumab subgroup (N=108; 2-year PFS of 23.1% vs. 64.1%; log-rank p-value = 0.003). Conclusions: This explorative analysis from the multicentre ALLSTAR registry demonstrates the predictive potential of MLR and dMLR for LRC and PFS. Since these blood biomarkers can be easily derived from standard pre-treatment bloodsamples they can be readily integrated in clinical routine. These blood biomarkers can be readily integrated in clinical routine since they are easily available.
Background: As demonstrated by the PACIFIC trial biomarker driven patient selection is crucial. While treatment based on programmed death ligand-1 (PD-L1) and mutational status have become routine, tests for biomarkers available from pre-therapeutic blood samples are currently a topic of scientific interest. Methods: This analysis was conducted in patients from the ALLSTAR RWD study, which is a nationwide, prospective registry for inoperable non-small cell lung cancer (NSCLC) stage III. Patients were amenable if they had a full routine pre-treatment blood sample, from which the following biomarkers were extracted: neutrophil-to-lymphocyte ratio (NLR), derived neutrophil-to-lymphocyte ratio (dNLR), platelet-to-lymphocyte ratio (PLR), monocyte-to-lymphocyte ratio (MLR), derived monocyte-to-lymphocyte ratio (dMLR) and lactate dehydrogenase (LDH) levels. The intention was to find a cutoff for each of these biomarkers to predict loco-regional control (LRC), progression free survival (PFS) and overall survival (OS). Results: MLR and dMLR demonstrated their predictive potential with cutoff values of 0.665 and 0.945, respectively. Stratifying the whole cohort by means of these cutoffs demonstrated significantly better loco-regional control for patients below threshold both in the whole cohort (N=175; 55.7% vs. 75.5%; p-value=0.018) and the Durvalumab subgroup (N=106; 57.5% vs. 77.3%; p-value = 0.030). Similar findings were observed for PFS in the whole cohort (N=175; 20.5% vs. 56.1%; p-value p<0.001) and the Durvalumab subgroup (N=106; 31.2% vs. 64.6%, p-value <0.001). dMLR could also significantly predict PFS (N=173; 17.4% vs. 56.3%; p-value <0.001), which was corroborated in the Durvalumab subgroup (N=108; 23.1% vs. 64.1%; p-value = 0.003). Conclusions: This explorative analysis demonstrates the predictive potential of MLR and dMLR for LRC and PFS. These blood biomarkers can be readily integrated in clinical routine since they are easily available.
Comment 3. In Table 1, it is recommended to add the group names in the table header. In addition, since no patients had unknown N-stage, this category could be removed.
Response 3. We agree with reviewer 1 and provide the modified table 1 below.
|
Patients |
||||
|
|
All |
Durvalumab |
p-value |
|
|
N=183 (%) |
N=112 (%) |
|||
|
Sex |
Male |
111 (60.7) |
70 (62.5) |
0.990 |
|
Female |
72 (39.3) |
42 (37.5) |
||
|
Age |
Median |
67.3 |
67.4 |
0.899 |
|
Range |
36.4-90.7 |
40.6-83.9 |
||
|
Smoking |
Never |
13 (7.1) |
10 (8.9) |
0.703 |
|
Ex |
104 (56.8) |
63 (56.3) |
||
|
Current |
66 (36.1) |
39 (34.8) |
||
|
ECOG |
0-1 |
171(93.4) |
106 (94.6) |
0.375 |
|
2-3 |
12 (6.6) |
6 (5.4) |
||
|
Histology |
AC |
95 (51.9) |
61 (54.5) |
0.559 |
|
SCC |
77 (42.1) |
47 (42.0) |
||
|
NOS |
11 (6.0) |
4 (3.5) |
||
|
PD-L1 |
<1% |
40 (21.9) |
12 (10.7) |
0.097 |
|
>1% |
128 (69.9) |
93 (83.0) |
||
|
Unknown |
15 (8.2) |
7 (6.3) |
||
|
T-stage |
Tis |
2 (1.1) |
1 (0.9) |
0.475 |
|
1 |
26 (14.2) |
18 (16.1) |
||
|
2 |
21 (11.5) |
13 (11.6) |
||
|
3 |
57 (31.1) |
38 (33.9) |
||
|
4 |
76 (41.6) |
41 (36.6) |
||
|
Unknown |
1 (0.5) |
1 (0.9) |
||
|
N-stage |
0 |
14 (7.7) |
8 (7.1) |
0.883 |
|
1 |
16 (8.7) |
7 (6.3) |
||
|
2 |
112 (61.2) |
76 (67.9) |
||
|
3 |
41 (22.4) |
21 (18.7) |
||
|
M-stage |
0 |
183 (100.0) |
112 (100.0) |
1.0 |
|
1 |
0 (0.0) |
0.0 (0.0) |
||
|
UICC |
IIIa |
64 (34.9) |
41 (36.6) |
0.631 |
|
IIIb |
83 (45.4) |
52 (46.4) |
||
|
IIIc |
36 (19.7) |
19 (17.0) |
||
Comment 4. In Table 1, it is unclear why patients with M-stage 1 were classified as stage III.
Response 4. We apologize for this typing error. All of the patients included in this study were UICC stage III, which means that they are – by definition – M0. Please find the corrected table below.
|
Patients |
||||
|
|
All |
Durvalumab |
p-value |
|
|
N=183 (%) |
N=112 (%) |
|||
|
Sex |
Male |
111 (60.7) |
70 (62.5) |
0.990 |
|
Female |
72 (39.3) |
42 (37.5) |
||
|
Age |
Median |
67.3 |
67.4 |
0.899 |
|
Range |
36.4-90.7 |
40.6-83.9 |
||
|
Smoking |
Never |
13 (7.1) |
10 (8.9) |
0.703 |
|
Ex |
104 (56.8) |
63 (56.3) |
||
|
Current |
66 (36.1) |
39 (34.8) |
||
|
ECOG |
0-1 |
171(93.4) |
106 (94.6) |
0.375 |
|
02.Mär |
12 (6.6) |
6 (5.4) |
||
|
Histology |
AC |
95 (51.9) |
61 (54.5) |
0.559 |
|
SCC |
77 (42.1) |
47 (42.0) |
||
|
NOS |
11 (6.0) |
4 (3.5) |
||
|
PD-L1 |
<1% |
40 (21.9) |
12 (10.7) |
0.097 |
|
>1% |
128 (69.9) |
93 (83.0) |
||
|
Unknown |
15 (8.2) |
7 (6.3) |
||
|
T-stage |
Tis |
2 (1.1) |
1 (0.9) |
0.475 |
|
1 |
26 (14.2) |
18 (16.1) |
||
|
2 |
21 (11.5) |
13 (11.6) |
||
|
3 |
57 (31.1) |
38 (33.9) |
||
|
4 |
76 (41.6) |
41 (36.6) |
||
|
Unknown |
1 (0.5) |
1 (0.9) |
||
|
N-stage |
0 |
14 (7.7) |
8 (7.1) |
0.883 |
|
1 |
16 (8.7) |
7 (6.3) |
||
|
2 |
112 (61.2) |
76 (67.9) |
||
|
3 |
41 (22.4) |
21 (18.7) |
||
|
M-stage |
0 |
183 (100.0) |
112 (100.0) |
1.0 |
|
1 |
0 (0.0) |
0.0 (0.0) |
||
|
UICC |
IIIa |
64 (34.9) |
41 (36.6) |
0.631 |
|
IIIb |
83 (45.4) |
52 (46.4) |
||
|
IIIc |
36 (19.7) |
19 (17.0) |
||
Comment 5. Based on the p-values in Figure 2c and 2d, the results do not seem statistically significant. The figure legends should avoid overly definitive statements.
Response 5. On re-reading these figure legends we have to admit that the choice of vocabulary might have been too exaggerated. Therefore we mitigated the legends as follows:
- c) (…). Patients with dMLR above cutoff showed a tendency towards inferior LRC, which did not reach statistical significance (N = 173; log-rank p-value 0.054).
- d) (…). Patients with dMLR above cutoff had nominally inferior LRC, which did not reach statistical significance (N = 108; log-rank p-value 0.209).
Comment 6. The abbreviations “UVA” and “MVA” lack full spelling. Based on the tables, these appear to refer to “univariate analysis” and “multivariate analysis.” Please provide the full terms at their first mention.
Response 6. As suggested by the reviewer we inserted the full terms when they are mentioned for the first time in the manuscript text (please see first line of the revised section 3.4 Multivariate analyses)
When MLR was tested in the univariate (UVA) and multivariate model (MVA) together with baseline and treatment characteristics (N = 183), (…).
Comment 7. The descriptions in Tables 3 and 4 do not fully correspond to those in the main text. It is recommended to present the results in the form of HR (95% CI) in the tables.
Response 7. We apologize for being inaccurate in this respect. Please find below the modified tables and the amended version of the text that we inserted in the manuscript (discussion section 3.4).
Table 3a. Locoregional control (LRC): Multivariate analysis for MLR and clinical characteristics revealed histology as the only significant parameter that impacted LRC in the whole cohort (non-SCC (106/183; 58%) versus SCC (77/183; 42%); p-value = 0.002; HR 2.09; 95%-CI 1.31-3.32) as well as in the Durvalumab subgroup (SCC (47/112; 42%) versus non-SCC (65/112; 58%); p-value < 0.001; HR 3.22; 95%-CI 1.70-6.07).
|
Baseline characteristics |
||||
|
|
N = 183 |
N = 112 |
||
|
UVA |
MVA |
UVA |
MVA |
|
|
Age (continuous) |
0.285 |
n.s. |
0.887 |
n.s. |
|
Sex (male vs female) |
0.629 |
n.s. |
0.859 |
n.s. |
|
Histology (non-SCC vs SCC) |
0.002 |
0.002; HR 2.09; 95%CI 1.31-3.32 |
<0.001 |
<0.001; HR 3.22; 95%CI 1.70-6.07 |
|
UICC (IIIa vs IIIb/c) |
0.802 |
n.s. |
0.431 |
n.s. |
|
MLR (</> 0.665) |
0.033 |
n.s. |
0.058 |
n.s. |
Table 3b. Locoregional control (LRC): Multivariate analysis for MLR and treatment characteristics revealed MLR as the only significant parameter that impacted LRC in the whole cohort (MLR <0.665 (N=146/183; 80%) versus p-value = 0.046; HR 2.07;95%-CI 1.01-4.22) with a strong trend in the Durvalumab group (univariate analysis: p-value 0.070).
|
Treatment characteristics |
||||
|
|
N = 183 |
N = 112 |
||
|
UVA |
MVA |
UVA |
MVA |
|
|
GTVTumour (continuous) |
0.124 |
n.s. |
0.672 |
n.s. |
|
GTVLymphnodes (continuous) |
0.165 |
n.s. |
0.064 |
n.s. |
|
EQD2Tumour (</> 66 Gy) |
0.218 |
n.s. |
0.778 |
n.s. |
|
EQD2Lymphnodes (</> 66 Gy) |
0.235 |
n.s. |
0.323 |
n.s. |
|
cCRT vs sCRT |
0.858 |
n.s. |
0.774 |
n.s. |
|
MLR (</> 0.665) |
0.041 |
0.046; HR 2.07; 95%CI 1.01-4.22 |
0.070 |
n.s. |
Table 3c. Locoregional control (LRC): Multivariate analysis for dMLR and clinical characteristics revealed histology as the only significant parameter that impacted LRC in the whole cohort (non-SCC (106/183; 58%) versus SCC (77/183; 42%); p-value <0.001; HR 2.71; 95%-CI 1.36-3.46) and the Durvalumab subgroup (non-SCC (65/112; 58%) versus SCC (47/112; 42%); p-value <0.001; HR 3.09; 95%-CI 1.68-5.72).
|
Baseline characteristics |
||||
|
|
N = 183 |
N = 112 |
||
|
UVA |
MVA |
UVA |
MVA |
|
|
Age (continuous) |
0.506 |
n.s. |
0.854 |
n.s. |
|
Sex (male vs female) |
0.608 |
n.s. |
0.977 |
n.s. |
|
Histology (non-SCC vs SCC) |
<0.001 |
<0.001; HR 2.71; 95%CI 1.36-3.46 |
<0.001 |
<0.001; HR 3.09; 95%CI 1.68-5.72 |
|
UICC (IIIa vs IIIb/c) |
0.856 |
n.s. |
0.374 |
n.s. |
|
dMLR (</>0.945) |
0.055 |
n.s. |
0.212 |
n.s. |
Table 3d. Locoregional control (LRC): Multivariate analysis for dMLR and treatment characteristics revealed no statistically significant parameter that impacted LRC, with a strong trend only for dMLR in the whole cohort (univariate analysis: p-value 0.057).
|
Treatment characteristics |
||||
|
|
N = 183 |
N = 112 |
||
|
UVA |
MVA |
UVA |
MVA |
|
|
GTVTumour (continuous) |
0.103 |
n.s. |
0.563 |
n.s. |
|
GTVLymphnodes (continuous) |
0.067 |
n.s. |
0.058 |
n.s. |
|
EQD2Tumour (</> 66 Gy) |
0.196 |
n.s. |
0.731 |
n.s. |
|
EQD2Lymphnodes (</> 66 Gy) |
0.290 |
n.s. |
0.357 |
n.s. |
|
cCRT vs sCRT |
0.857 |
n.s. |
0.739 |
n.s. |
|
dMLR (</>0.945) |
0.057 |
n.s. |
0.217 |
n.s. |
Table 4a. Progression free survival (PFS): In the whole cohort multivariate analysis for MLR and clinical characteristics showed that histology (non-SCC (106/183;58%) versus SCC (77/183;42%); p-value = 0.023; HR 1.50; 95%-CI 1.06-2.12), UICC (IIIa (64/183; 35%) vs IIIb/c (119/183; 65%); p-value = 0.016; HR 1.44; 95%-CI 1.07-1.94) and MLR (>0.665 (N=29/183; 16%) versus <0.665 (146/183; 80%); p-value = 0.002; HR 2.23; 95%-CI 1.34-3.71) significantly impacted PFS. This finding was corroborated in the Durvalumab subgroup with histology (non-SCC (65/112; 58%) versus SCC (47/112; 42%); p-value = 0.009; HR 1.90; 95%-CI 1.17-3.09) and MLR (<0.665 (N=89/108; 84%) versus >0.665 (N=17/108; 16%); p-value = 0.012; HR 2.38; 95%-CI 1.21-4.66) as significant parameters.
|
Baseline characteristics |
||||
|
|
N = 183 |
N = 112 |
||
|
|
UVA |
MVA |
UVA |
MVA |
|
Age (continuous) |
0.316 |
n.s |
0.760 |
n.s. |
|
Sex (male vs female) |
0.772 |
n.s |
0.791 |
n.s. |
|
Histology (non-SCC vs SCC) |
0.010 |
0.023; HR 1.50; 95%CI 1.06-2.12 |
0.008 |
0.009; HR 1.90; 95%CI 1.17-3.09 |
|
UICC (IIIa vs IIIb/c) |
0.031 |
0.016; HR 1.44; 95%CI 1.07-1.94 |
0.083 |
n.s. |
|
MLR (</> 0.665) |
<0.001 |
0.002; HR 2.23; 95%CI 1.34-3.71 |
0.010 |
0.012; HR 2.38; 95%CI 1.21-4.66 |
Table 4b. Progression free survival (PFS): Multivariate analysis for MLR and treatment characteristics revealed MLR as the only parameter with significant impact on PFS both in the whole cohort (<0.665 (146/183; 80%) versus >0.665 (N=29/183; 16%); p-value < 0.001; HR 2.23; 95%-CI 1.43-3.88) and the Durvalumab subgroup (<0.665 (N=89/108; 84%) versus >0.665 (N=17/108; 16%); p-value = 0.012; HR 2.38; 95%-CI 1.21-4.66).
|
Treatment characteristics |
||||
|
|
N = 183 |
N = 112 |
||
|
UVA |
MVA |
UVA |
MVA |
|
|
GTVTumour (continuous) |
0.095 |
n.s |
0.290 |
n.s. |
|
GTVLymphnodes (continuous) |
0.045 |
n.s. |
0.920 |
n.s. |
|
cCRT vs sCRT |
0.104 |
n.s. |
0.283 |
n.s. |
|
ICI treatment (yes vs no) |
0.873 |
n.s. |
n.a. |
n.a |
|
MLR (</> 0.665) |
<0.001 |
<0.001; HR 2.23; 95%CI 1.43-3.88 |
0.010 |
0.012; HR 2.38; 95%CI 1.21-4.66 |
Table 4c. Progression free survival (PFS): Multivariate analysis for dMLR and clinical characteristics revealed that dMLR was the only significant factor to impact PFS in the whole cohort (<0.945 (N=147/173; 85%) versus dMLR>0.945 (N=26/173;15%); p-value = 0.001; HR 2.35; 95%-CI: 1.40-3.95) as well as in the Durvalumab subgroup (<0.945 (N=90/108; 83%) versus >0.945 (N=18/108; 17%); p-value = 0.005; HR 2.54; 95%-CI 1.33-4.85).
|
Baseline characteristics |
||||
|
|
N = 183 |
N = 112 |
||
|
UVA |
MVA |
UVA |
MVA |
|
|
Age (continuous) |
0.398 |
n.s. |
0.880 |
n.s. |
|
Sex (male vs female) |
0.788 |
n.s. |
0.781 |
n.s. |
|
Histology (non-SCC vs SCC) |
0.030 |
n.s. |
0.021 |
n.s. |
|
UICC (IIIa vs IIIb/c) |
0.091 |
n.s. |
0.099 |
n.s. |
|
dMLR (</>0.945) |
<0.001 |
0.001; HR 2.35; 95%CI 1.40-3.95 |
0.004 |
0.005; HR 2.54; 95%CI 1.33-4.85 |
Table 4d. Progression free survival (PFS): Multivariate analysis for dMLR and clinical characteristics revealed that dMLR was the only significant factor to impact PFS in the whole cohort (<0.945 (N=147/173; 85%) versus >0.945 (N=26/173;15%); p-value <0.001; HR 2.35; 95%-CI: 1.40-3.95) as well as in the Durvalumab subgroup (<0.945 (N=90/108; 83%) versus >0.945 (N=18/108; 17%); p-value = 0.004; HR 2.64; 95%-CI 1.38-5.07).
|
Treatment characteristics |
||||
|
|
N = 183 |
N = 112 |
||
|
UVA |
MVA |
UVA |
MVA |
|
|
GTVTumour (continuous) |
0.080 |
n.s. |
0.298 |
n.s. |
|
GTVLymphnodes (continuous) |
0.632 |
n.s. |
0.955 |
n.s. |
|
cCRT vs sCRT |
0.056 |
n.s. |
0.639 |
n.s. |
|
ICI treatment (yes vs no) |
0.884 |
n.s. |
0.877 |
n.s. |
|
dMLR (</>0.945) |
<0.001 |
0.001; HR 2.35; 95%CI 1.40-3.95 |
0.002 |
0.004; HR 2.64; 95%CI 1.38-5.07 |
(…)
When MLR was tested in the univariate (UVA) and multivariate model (MVA) together with baseline and treatment characteristics (N = 183), the following variables had a significant impact on LRC: histology (p-value = 0.002; HR 2.09; 95%-CI 1.31-3.32; table 3a), MLR >0.665 (p-value = 0.046; HR 2.07;95%-CI 1.01-4.22; table 3b). This finding could be partially corroborated in the Durvalumab subgroup with SCC as an independent predictor for worse LRC (p<0.001; HR 3.22;95% CI 1.70-6.07; table 3a). A separate analysis with the same baseline and treatment characteristics combined with dMLR instead of MLR revealed that histology was the only predictive parameter for LRC both in the whole cohort (p-value <0.001; HR 2.71; 95%-CI 1.36-3.46; table 3c) and the Durvalumab subgroup (p-value <0.001; HR 3.09; 95%-CI 1.68-5.72; table 3c).
When MLR was tested in the MVA model together with baseline and treatment characteristics (N = 183), the following variables had a significant impact on PFS: histology (p-value = 0.023; HR 1.50; 95%-CI 1.06-2.12; table 4a), UICC (p-value = 0.016; HR 1.44; 95%-CI 1.07-1.94; table 4a) and MLR (table 4a: p-value = 0.002; HR 2.23; 95%-CI 1.34-3.71; table 4b: p-value < 0.001; HR 2.23; 95%-CI 1.43-3.88). In the Durvalumab subgroup risk factors for a poorer PFS were histology (p-value =0.009; HR 1.90; 95%-CI 1.17-3.09; table 4a) and MLR (p-value = 0.012; HR 2.38; 95%-CI 1.21-4.66; table 4a and 4b). A higher dMLR was the only significant factor to be associated with poorer PFS in the whole cohort (p-value = 0.001; HR 2.35; 95%-CI: 1.40-3.95; table 4c and table 4d) as well as in the Durvalumab subgroup (table 4c: p-value = 0.005; HR 2.54; 95%-CI 1.33-4.85; table 4d: p-value = 0.004; HR 2.64; 95%-CI 1.38-5.07).
Comment 8. The analyses in the manuscript are limited to PFS and LRC. It is recommended to discuss the potential value regarding OS in the Discussion section.
Response 8. We thank reviewer 1 for bringing up this important issue. In order to adequately answer this question we added supplementary table 1 (results section 3.2; see also below). With OS as an endpoint, we found a significant threshold for dMLR (very last line of the table) but not for MLR. The threshold for dMLR (last line from the bottom: 0.6) however differed from the one for LRC (third line from the bottom: 0.945) and PFS (last but one line of the table: 0.945). As we intended to use the same threshold in both the whole cohort and the Durvalumab subgroup, we decided to use 0.945, which seems to be a robust cutoff for our data. This value however, was not significant with respect to OS. Additionally, we found significant cutoffs for NLR, dNLR and LDH with OS as an endpoint. These values however were not reproducible in the Durvalumab cohort or with other endpoints such as LRC or PFS so that we did not proceed in analysing them further. This reflects the main issue with this type of research: various endpoints lead to different cutoff thresholds for the same blood parameter. To address this, we added the following sentences in the discussion section (line 343):
As for OS, we were unable to detect a cutoff value for any of the analyzed blood biomarkers that was stringently correlated with this endpoint both in the whole cohort and the Durvalumab subgroup (see supplementary table 1).
Supplementary table 1. Panel of all tested biomarkers.
|
Biomarker |
|||||||
|
N = 183 |
N =112 |
||||||
|
Threshold interval (p<0.05) |
Threshold |
P-value for threshold |
Threshold interval (p<0.05) |
Threshold |
P-value for threshold |
||
|
NLR |
LRC |
none |
none |
none |
none |
none |
none |
|
PFS |
2.842* |
2 842 |
0.043 |
12.75-13.27 |
12.75 |
0.029 |
|
|
OS |
2.895-2.906 |
2 906 |
0.031 |
24.088* |
24 088 |
0.017 |
|
|
MLR |
LRC |
0.663-0.696 |
0.665 |
0.018 |
0.663-0696 |
0.665 |
0.030 |
|
PFS |
0.450-0.786 |
0.665 |
<0.001 |
0.420 - 0.677 |
0.665 |
0.004 |
|
|
OS |
none |
none |
none |
none |
none |
none |
|
|
PLR |
LRC |
79.091* |
79 091 |
0.013 |
none |
none |
none |
|
PFS |
71.777* |
71 777 |
0.050 |
79.091* |
79.091 |
0.046 |
|
|
OS |
none |
none |
none |
79.091 or 597.895** |
79.091 |
0,009 |
|
|
dNLR |
LRC |
3.876-4.293 |
4 |
0.037 |
none |
none |
none |
|
PFS |
none |
none |
none |
none |
none |
none |
|
|
OS |
4.255-4.364 |
4.293 |
0.025 |
3.815-3.876 |
3.815 |
0.030 |
|
|
LDH |
LRC |
152-155 |
154 |
0.012 |
146 - 153 |
153 |
0.010 |
|
PFS |
108* |
108 |
0.042 |
177* |
177 |
0.037 |
|
|
OS |
324-331 |
327 |
<0.001 |
194* |
194 |
0.039 |
|
|
dMLR |
LRC |
0.945-0.987 |
0.945 |
0.054 |
1.173* |
0.945 |
0.209 |
|
PFS |
0.58-1.077 |
0.945 |
<0.001 |
0.769-1,033 |
0.945 |
0.003 |
|
|
OS |
0.58-0.688 |
0.6 |
0.015 |
none |
none |
none |
|
* Only one significant value
** Two independent thresholds with skewed group distribution.

Reviewer 2 Report
Comments and Suggestions for Authors
- The author should provide the ethical approval number for this study.
- Most of the baseline features look balanced, but, still, please show the P values of the differences in baseline parameters in Table 1.
- Similarly, please also show the P values in Table 2.
- The Durvalumab subgroup does not belong to the whole cohort. So, this terminology will make misunderstandings for readers. Please optimize them.
- Please revise all tables representing multivariate analysis results. For each variable, include the number of cases and percentages of each category, along with hazard ratios (HR) and 95% confidence intervals (CI). Additionally, please add the corresponding univariate analysis results for comparison.
- What about the significance of different combinations of blood parameters?
Author Response
First of all, we would like to thank the reviewer for his/her in-depth analysis of our manuscript. We tried to answer the issues raised to the best of our knowlegde. Please find our point-by-point answers below (insertions in the manuscript text are written in blue).
Reviewer 2
Comment 1. The author should provide the ethical approval number for this study.
Response 1. We apologize for the oversight. The study was approved by the ethics board of the federal state of Salzburg under the following approval number: Ethikkommission für das Bundesland Salzburg 1002/2019. Hence we added the following sentence under 2.1 Patients:
Approval by the ethics committee of the federal state of Salzburg was obtained on the 20th of March 2020 (approval number: 1002/2019).
Comment 2. Most of the baseline features look balanced, but, still, please show the P values of the differences in baseline parameters in Table 1.
Response 2. Yes indeed, the baseline characteristics are fairly balanced, which is corroborated by the non-parametric comparison (Mann-Whitney-U) between groups. As suggested by the reviewer, we added the p-values in a separate column and replaced the original table by the one below.
|
Patients |
||||
|
|
All |
Durvalumab |
p-value |
|
|
N=183 (%) |
N=112 (%) |
|||
|
Sex |
Male |
111 (60.7) |
70 (62.5) |
0.990 |
|
Female |
72 (39.3) |
42 (37.5) |
||
|
Age |
Median |
67.3 |
67.4 |
0.899 |
|
Range |
36.4-90.7 |
40.6-83.9 |
||
|
Smoking |
Never |
13 (7.1) |
10 (8.9) |
0.703 |
|
Ex |
104 (56.8) |
63 (56.3) |
||
|
Current |
66 (36.1) |
39 (34.8) |
||
|
ECOG |
0-1 |
171(93.4) |
106 (94.6) |
0.375 |
|
02.Mär |
12 (6.6) |
6 (5.4) |
||
|
Histology |
AC |
95 (51.9) |
61 (54.5) |
0.559 |
|
SCC |
77 (42.1) |
47 (42.0) |
||
|
NOS |
11 (6.0) |
4 (3.5) |
||
|
PD-L1 |
<1% |
40 (21.9) |
12 (10.7) |
0.097 |
|
>1% |
128 (69.9) |
93 (83.0) |
||
|
Unknown |
15 (8.2) |
7 (6.3) |
||
|
T-stage |
Tis |
2 (1.1) |
1 (0.9) |
0.475 |
|
1 |
26 (14.2) |
18 (16.1) |
||
|
2 |
21 (11.5) |
13 (11.6) |
||
|
3 |
57 (31.1) |
38 (33.9) |
||
|
4 |
76 (41.6) |
41 (36.6) |
||
|
Unknown |
1 (0.5) |
1 (0.9) |
||
|
N-stage |
0 |
14 (7.7) |
8 (7.1) |
0.883 |
|
1 |
16 (8.7) |
7 (6.3) |
||
|
2 |
112 (61.2) |
76 (67.9) |
||
|
3 |
41 (22.4) |
21 (18.7) |
||
|
M-stage |
0 |
183 (100.0) |
112 (100.0) |
1.0 |
|
1 |
0 (0.0) |
0.0 (0.0) |
||
|
UICC |
IIIa |
64 (34.9) |
41 (36.6) |
0.631 |
|
IIIb |
83 (45.4) |
52 (46.4) |
||
|
IIIc |
36 (19.7) |
19 (17.0) |
||
Comment 3. Similarly, please also show the P values in Table 2.
Response 3. As requested by the reviewer, we also added the p-values in table 2. With the exception of immune checkpoint inhibition, which is plausible since all patients in the Durvalumab group were treated with Durvalumab, treatment characteristics are also balanced. This version of table 2 replaces the original one.
|
Treatment |
||||
|
|
All N=183 (%) |
Durvalumab N=112 (%) |
p-value |
|
|
Treatment sequence |
cCRT |
46 (25.1) |
30 (26.8) |
0.384 |
|
sCRT |
126 (68.9) |
81 (72.3) |
||
|
RIT |
11 (6.0) |
1 (0.9) |
||
|
Immune Checkpoint Inhibitors |
Durvalumab |
112 (61.2) |
112 (100.0) |
<0.001 |
|
Other ICI* |
16 (8.7) |
0 (0.0) |
||
|
No ICI |
55 (30.1) |
0 (0.0) |
||
|
RT-Technique |
VMAT/IMRT |
178 (97.3) |
109 (97.3) |
0.659 |
|
3D |
5 (2.7) |
3 (2.7) |
||
|
Tumour |
EQD2 median |
65.0 |
67.1 |
0.051 |
|
EQD2 range |
0.0-100.0 |
0.0-100.0 |
||
|
GTV median |
48.4 |
44.4 |
0.242 |
|
|
GTV range |
0.0-784.1 |
0.0-483.8 |
||
|
Lymph nodes |
EQD2 median |
57.3 |
57.3 |
0.309 |
|
EQD2 range |
0.0-81.3 |
0.0-70.0 |
||
|
GTV median |
29.0 |
26.0 |
0.364 |
|
|
GTV range |
0.0-473.0 |
0.0-293.3 |
||
|
Elective nodal irradiation |
EQD2 median |
32.5 |
48.8 |
0.832 |
|
EQD2 range |
32.5-60.0 |
32.5-60.0 |
||
|
GTV median |
90.0 |
79.0 |
0.523 |
|
|
GTV range |
0.0-429.0 |
0.0-429.0 |
||
Comment 4. The Durvalumab subgroup does not belong to the whole cohort. So, this terminology will make misunderstandings for readers. Please optimize them.
Response 4. Many thanks for this remark. ALLSTAR, which is a nationwide prospective registry, included NSCLC stage III patients regardless of whether they were treated with immunotherapy or not. Hence the whole cohort consists of 183 patients. In order to shed light on the most interesting subgroup of these 183 patients, i.e. those who received Durvalumab (N=112), we analyzed the Durvalumab patients – which form part of the whole cohort – separately. In order to clarify this we inserted the following sentence at the end of section 2.1.
All analyses were conducted first in the whole cohort (N = 183) and in a second step in a subpopulation of these patients who were treated with Durvalumab (N = 112).
Comment 5. Please revise all tables representing multivariate analysis results. For each variable, include the number of cases and percentages of each category, along with hazard ratios (HR) and 95% confidence intervals (CI). Additionally, please add the corresponding univariate analysis results for comparison.
Response 5. According to the reviewer’s suggestion we modified all the tables including multivariate analyses. The results of the univariate analyses can be found in the second and fourth column of each table.
Table 3a. Locoregional control (LRC): Multivariate analysis for MLR and clinical characteristics revealed histology as the only significant parameter that impacted LRC in the whole cohort (non-SCC (106/183; 58%) versus SCC (77/183; 42%); p-value = 0.002; HR 2.09; 95%-CI 1.31-3.32) as well as in the Durvalumab subgroup (SCC (47/112; 42%) versus non-SCC (65/112; 58%); p-value < 0.001; HR 3.22; 95%-CI 1.70-6.07).
|
Baseline characteristics |
||||
|
|
N = 183 |
N = 112 |
||
|
UVA |
MVA |
UVA |
MVA |
|
|
Age (continuous) |
0.285 |
n.s. |
0.887 |
n.s. |
|
Sex (male vs female) |
0.629 |
n.s. |
0.859 |
n.s. |
|
Histology (non-SCC vs SCC) |
0.002 |
0.002; HR 2.09; 95%CI 1.31-3.32 |
<0.001 |
<0.001; HR 3.22; 95%CI 1.70-6.07 |
|
UICC (IIIa vs IIIb/c) |
0.802 |
n.s. |
0.431 |
n.s. |
|
MLR (</> 0.665) |
0.033 |
n.s. |
0.058 |
n.s. |
Table 3b. Locoregional control (LRC): Multivariate analysis for MLR and treatment characteristics revealed MLR as the only significant parameter that impacted LRC in the whole cohort (MLR <0.665 (N=146/183; 80%) versus p-value = 0.046; HR 2.07;95%-CI 1.01-4.22) with a strong trend only in the Durvalumab group (univariate analysis: p-value 0.070).
|
Treatment characteristics |
||||
|
|
N = 183 |
N = 112 |
||
|
UVA |
MVA |
UVA |
MVA |
|
|
GTVTumour (continuous) |
0.124 |
n.s. |
0.672 |
n.s. |
|
GTVLymphnodes (continuous) |
0.165 |
n.s. |
0.064 |
n.s. |
|
EQD2Tumour (</> 66 Gy) |
0.218 |
n.s. |
0.778 |
n.s. |
|
EQD2Lymphnodes (</> 66 Gy) |
0.235 |
n.s. |
0.323 |
n.s. |
|
cCRT vs sCRT |
0.858 |
n.s. |
0.774 |
n.s. |
|
MLR (</> 0.665) |
0.041 |
0.046; HR 2.07; 95%CI 1.01-4.22 |
0.070 |
n.s. |
Table 3c. Locoregional control (LRC): Multivariate analysis for dMLR and clinical characteristics revealed histology as the only significant parameter that impacted LRC in the whole cohort (non-SCC (106/183; 58%) versus SCC (77/183; 42%); p-value <0.001; HR 2.71; 95%-CI 1.36-3.46) and the Durvalumab subgroup (non-SCC (65/112; 58%) versus SCC (47/112; 42%); p-value <0.001; HR 3.09; 95%-CI 1.68-5.72).
|
Baseline characteristics |
||||
|
|
N = 183 |
N = 112 |
||
|
UVA |
MVA |
UVA |
MVA |
|
|
Age (continuous) |
0.506 |
n.s. |
0.854 |
n.s. |
|
Sex (male vs female) |
0.608 |
n.s. |
0.977 |
n.s. |
|
Histology (non-SCC vs SCC) |
<0.001 |
<0.001; HR 2.71; 95%CI 1.36-3.46 |
<0.001 |
<0.001; HR 3.09; 95%CI 1.68-5.72 |
|
UICC (IIIa vs IIIb/c) |
0.856 |
n.s. |
0.374 |
n.s. |
|
dMLR (</>0.945) |
0.055 |
n.s. |
0.212 |
n.s. |
Table 3d. Locoregional control (LRC): Multivariate analysis for dMLR and treatment characteristics revealed no statistically significant parameter that impacted LRC, with a strong trend only for dMLR in the whole cohort (univariate analysis: p-value 0.057).
|
Treatment characteristics |
||||
|
|
N = 183 |
N = 112 |
||
|
UVA |
MVA |
UVA |
MVA |
|
|
GTVTumour (continuous) |
0.103 |
n.s. |
0.563 |
n.s. |
|
GTVLymphnodes (continuous) |
0.067 |
n.s. |
0.058 |
n.s. |
|
EQD2Tumour (</> 66 Gy) |
0.196 |
n.s. |
0.731 |
n.s. |
|
EQD2Lymphnodes (</> 66 Gy) |
0.290 |
n.s. |
0.357 |
n.s. |
|
cCRT vs sCRT |
0.857 |
n.s. |
0.739 |
n.s. |
|
dMLR (</>0.945) |
0.057 |
n.s. |
0.217 |
n.s. |
Table 4a. Progression free survival (PFS): In the whole cohort multivariate analysis for MLR and clinical characteristics showed that histology (non-SCC (106/183;58%) versus SCC (77/183;42%); p-value = 0.023; HR 1.50; 95%-CI 1.06-2.12), UICC (IIIa (64/183; 35%) vs IIIb/c (119/183; 65%); p-value = 0.016; HR 1.44; 95%-CI 1.07-1.94) and MLR (>0.665 (N=29/183; 16%) versus <0.665 (146/183; 80%); p-value = 0.002; HR 2.23; 95%-CI 1.34-3.71) significantly impacted PFS. This finding was corroborated in the Durvalumab subgroup with histology (non-SCC (65/112; 58%) versus SCC (47/112; 42%); p-value = 0.009; HR 1.90; 95%-CI 1.17-3.09) and MLR (<0.665 (N=89/108; 84%) versus >0.665 (N=17/108; 16%); p-value = 0.012; HR 2.38; 95%-CI 1.21-4.66) as significant parameters.
|
Baseline characteristics |
||||
|
|
N = 183 |
N = 112 |
||
|
|
UVA |
MVA |
UVA |
MVA |
|
Age (continuous) |
0.316 |
n.s |
0.760 |
n.s. |
|
Sex (male vs female) |
0.772 |
n.s |
0.791 |
n.s. |
|
Histology (non-SCC vs SCC) |
0.010 |
0.023; HR 1.50; 95%CI 1.06-2.12 |
0.008 |
0.009; HR 1.90; 95%CI 1.17-3.09 |
|
UICC (IIIa vs IIIb/c) |
0.031 |
0.016; HR 1.44; 95%CI 1.07-1.94 |
0.083 |
n.s. |
|
MLR (</> 0.665) |
<0.001 |
0.002; HR 2.23; 95%CI 1.34-3.71 |
0.010 |
0.012; HR 2.38; 95%CI 1.21-4.66 |
Table 4b. Progression free survival (PFS): Multivariate analysis for MLR and treatment characteristics revealed MLR as the only parameter with significant impact on PFS both in the whole cohort (<0.665 (146/183; 80%) versus >0.665 (N=29/183; 16%); p-value < 0.001; HR 2.23; 95%-CI 1.43-3.88) and the Durvalumab subgroup (<0.665 (N=89/108; 84%) versus >0.665 (N=17/108; 16%); p-value = 0.012; HR 2.38; 95%-CI 1.21-4.66).
|
Treatment characteristics |
||||
|
|
N = 183 |
N = 112 |
||
|
UVA |
MVA |
UVA |
MVA |
|
|
GTVTumour (continuous) |
0.095 |
n.s |
0.290 |
n.s. |
|
GTVLymphnodes (continuous) |
0.045 |
n.s. |
0.920 |
n.s. |
|
cCRT vs sCRT |
0.104 |
n.s. |
0.283 |
n.s. |
|
ICI treatment (yes vs no) |
0.873 |
n.s. |
n.a. |
n.a |
|
MLR (</> 0.665) |
<0.001 |
<0.001; HR 2.23; 95%CI 1.43-3.88 |
0.010 |
0.012; HR 2.38; 95%CI 1.21-4.66 |
Table 4c. Progression free survival (PFS): Multivariate analysis for dMLR and clinical characteristics revealed that dMLR was the only significant factor to impact PFS in the whole cohort (<0.945 (N=147/173; 85%) versus dMLR>0.945 (N=26/173;15%); p-value = 0.001; HR 2.35; 95%-CI: 1.40-3.95) as well as in the Durvalumab subgroup (<0.945 (N=90/108; 83%) versus >0.945 (N=18/108; 17%); p-value = 0.005; HR 2.54; 95%-CI 1.33-4.85).
|
Baseline characteristics |
||||
|
|
N = 183 |
N = 112 |
||
|
UVA |
MVA |
UVA |
MVA |
|
|
Age (continuous) |
0.398 |
n.s. |
0.880 |
n.s. |
|
Sex (male vs female) |
0.788 |
n.s. |
0.781 |
n.s. |
|
Histology (non-SCC vs SCC) |
0.030 |
n.s. |
0.021 |
n.s. |
|
UICC (IIIa vs IIIb/c) |
0.091 |
n.s. |
0.099 |
n.s. |
|
dMLR (</>0.945) |
<0.001 |
0.001; HR 2.35; 95%CI 1.40-3.95 |
0.004 |
0.005; HR 2.54; 95%CI 1.33-4.85 |
Table 4d. Progression free survival (PFS): Multivariate analysis for dMLR and clinical characteristics revealed that dMLR was the only significant factor to impact PFS in the whole cohort (<0.945 (N=147/173; 85%) versus >0.945 (N=26/173;15%); p-value <0.001; HR 2.35; 95%-CI: 1.40-3.95) as well as in the Durvalumab subgroup (<0.945 (N=90/108; 83%) versus >0.945 (N=18/108; 17%); p-value = 0.004; HR 2.64; 95%-CI 1.38-5.07).
|
Treatment characteristics |
||||
|
|
N = 183 |
N = 112 |
||
|
UVA |
MVA |
UVA |
MVA |
|
|
GTVTumour (continuous) |
0.080 |
n.s. |
0.298 |
n.s. |
|
GTVLymphnodes (continuous) |
0.632 |
n.s. |
0.955 |
n.s. |
|
cCRT vs sCRT |
0.056 |
n.s. |
0.639 |
n.s. |
|
ICI treatment (yes vs no) |
0.884 |
n.s. |
0.877 |
n.s. |
|
dMLR (</>0.945) |
<0.001 |
0.001; HR 2.35; 95%CI 1.40-3.95 |
0.002 |
0.004; HR 2.64; 95%CI 1.38-5.07 |
Comment 6. What about the significance of different combinations of blood parameters?
Response 6. This is a very important question, whose answer is beyond the scope of the current project. At present we are planning to re-analyse the data in search of various combinations of blood markers, for example – based on the ground breaking work by Laura Mezquita (doi:10.1001/jamaoncol.2017.4771) – lung immune prognostic index, for which we have performed a very preliminary analysis thus far.

Round 2
Reviewer 1 Report
Comments and Suggestions for Authors
The authors have made excellent revisions; only one minor issue remains to be addressed: the number of decimal places should be standardized across the entire table. Even values such as 1.0 should be presented as 1.000.
Author Response
Comment 1. The authors have made excellent revisions; only one minor issue remains to be addressed: the number of decimal places should be standardized across the entire table. Even values such as 1.0 should be presented as 1.000.
Response 1. The reviewer is right. We corrected the decimals according to his/her suggestion (please find the table below). Changes in the manuscript text are in blue font.
Table 1
|
Patients |
||||
|
|
All |
Durvalumab |
p-value |
|
|
N=183 (%) |
N=112 (%) |
|||
|
Sex |
Male |
111 (60.7) |
70 (62.5) |
0.990 |
|
Female |
72 (39.3) |
42 (37.5) |
||
|
Age |
Median |
67.3 |
67.4 |
0.899 |
|
Range |
36.4-90.7 |
40.6-83.9 |
||
|
Smoking |
Never |
13 (7.1) |
10 (8.9) |
0.703 |
|
Ex |
104 (56.8) |
63 (56.3) |
||
|
Current |
66 (36.1) |
39 (34.8) |
||
|
ECOG |
0-1 |
171(93.4) |
106 (94.6) |
0.375 |
|
2-3 |
12 (6.6) |
6 (5.4) |
||
|
Histology |
AC |
95 (51.9) |
61 (54.5) |
0.559 |
|
SCC |
77 (42.1) |
47 (42.0) |
||
|
NOS |
11 (6.0) |
4 (3.5) |
||
|
PD-L1 |
<1% |
40 (21.9) |
12 (10.7) |
0.097 |
|
>1% |
128 (69.9) |
93 (83.0) |
||
|
Unknown |
15 (8.2) |
7 (6.3) |
||
|
T-stage |
Tis |
2 (1.1) |
1 (0.9) |
0.475 |
|
1 |
26 (14.2) |
18 (16.1) |
||
|
2 |
21 (11.5) |
13 (11.6) |
||
|
3 |
57 (31.1) |
38 (33.9) |
||
|
4 |
76 (41.6) |
41 (36.6) |
||
|
Unknown |
1 (0.5) |
1 (0.9) |
||
|
N-stage |
0 |
14 (7.7) |
8 (7.1) |
0.883 |
|
1 |
16 (8.7) |
7 (6.3) |
||
|
2 |
112 (61.2) |
76 (67.9) |
||
|
3 |
41 (22.4) |
21 (18.7) |
||
|
M-stage |
0 |
183 (100.0) |
112 (100.0) |
1.000 |
|
1 |
0 (0.0) |
0.0 (0.0) |
||
|
UICC |
IIIa |
64 (34.9) |
41 (36.6) |
0.631 |
|
IIIb |
83 (45.4) |
52 (46.4) |
||
|
IIIc |
36 (19.7) |
19 (17.0) |
||
Table 2
|
Patients |
||||
|
|
All |
Durvalumab |
p-value |
|
|
N=183 (%) |
N=112 (%) |
|||
|
Sex |
Male |
111 (60.7) |
70 (62.5) |
0.990 |
|
Female |
72 (39.3) |
42 (37.5) |
||
|
Age |
Median |
67.3 |
67.4 |
0.899 |
|
Range |
36.4-90.7 |
40.6-83.9 |
||
|
Smoking |
Never |
13 (7.1) |
10 (8.9) |
0.703 |
|
Ex |
104 (56.8) |
63 (56.3) |
||
|
Current |
66 (36.1) |
39 (34.8) |
||
|
ECOG |
0-1 |
171(93.4) |
106 (94.6) |
0.375 |
|
02.Mär |
12 (6.6) |
6 (5.4) |
||
|
Histology |
AC |
95 (51.9) |
61 (54.5) |
0.559 |
|
SCC |
77 (42.1) |
47 (42.0) |
||
|
NOS |
11 (6.0) |
4 (3.5) |
||
|
PD-L1 |
<1% |
40 (21.9) |
12 (10.7) |
0.097 |
|
>1% |
128 (69.9) |
93 (83.0) |
||
|
Unknown |
15 (8.2) |
7 (6.3) |
||
|
T-stage |
Tis |
2 (1.1) |
1 (0.9) |
0.475 |
|
1 |
26 (14.2) |
18 (16.1) |
||
|
2 |
21 (11.5) |
13 (11.6) |
||
|
3 |
57 (31.1) |
38 (33.9) |
||
|
4 |
76 (41.6) |
41 (36.6) |
||
|
Unknown |
1 (0.5) |
1 (0.9) |
||
|
N-stage |
0 |
14 (7.7) |
8 (7.1) |
0.883 |
|
1 |
16 (8.7) |
7 (6.3) |
||
|
2 |
112 (61.2) |
76 (67.9) |
||
|
3 |
41 (22.4) |
21 (18.7) |
||
|
M-stage |
0 |
183 (100.0) |
112 (100.0) |
1.000 |
|
1 |
0 (0.0) |
0.0 (0.0) |
||
|
UICC |
IIIa |
64 (34.9) |
41 (36.6) |
0.631 |
|
IIIb |
83 (45.4) |
52 (46.4) |
||
|
IIIc |
36 (19.7) |
19 (17.0) |
||

Reviewer 2 Report
Comments and Suggestions for Authors
Thanks to the authors for the quick response. However, there are still some issues about the tables. As a statistics-based research article, data presentation is critical. It’s recommended that the authors:
- Add a workflow chart to visually illustrate the experimental design, sample grouping, and analysis pipeline.
- Optimize the COX regression using a more standardized and reader-friendly format, such as the following layout:
|
Factors (# of cases) |
Univariate analysis |
|
Multivariate analysis |
||||||
|
P value |
HR |
95% CI |
P value |
HR |
95% CI |
||||
|
Upper |
Lower |
|
Upper |
Lower |
|||||
Author Response
We thank reviewer 2 for the useful comments on our revised manuscript. Please find out point-by-point answers below with insertions in the manuscript text in blue font.
Thanks to the authors for the quick response. However, there are still some issues about the tables. As a statistics-based research article, data presentation is critical. It’s recommended that the authors:
- Add a workflow chart to visually illustrate the experimental design, sample grouping, and analysis pipeline.
- Optimize the COX regression using a more standardized and reader-friendly format, such as the following layout:
Comment 1. Add a workflow chart to visually illustrate the experimental design, sample grouping, and analysis pipeline.
Response 1. We agree with reviewer 2 in as far as a workflow chart certainly helps the reader to better follow the analysis. Hence we added the chart below as supplemental file 1 and the following sentence at the end of section 2.4.
Please find the study overview in supplemental figure 1.
Scheme 1
*Biomarkers that were previously tested in another RWD study (Park PACIFIC-KR, reference 13)
^These biomarkers did not fulfill the inclusion criteria for further investigation defined in methods section 2.4.
Comment 2. Optimize the COX regression using a more standardized and reader-friendly format, such as the following layout:
|
Factors (# of cases) |
Univariate analysis |
|
Multivariate analysis |
||||||
|
P value |
HR |
95% CI |
|
P value |
HR |
95% CI |
|||
|
Upper |
Lower |
|
Upper |
Lower |
|||||
|
|
|
|
|
|
|
|
|
|
|
Response 2. We thank reviewer 2 for providing a template, which makes it easier for us to adequately answer this question. Please find the modified tables 3 and 4 below. Additionally we also modified the results section accordingly – please find the modified manuscript text at the end of the tables.
Table 3a. Locoregional control (LRC): Multivariate analysis for MLR and clinical characteristics revealed histology as the only significant parameter that impacted LRC in the whole cohort (p-value = 0.004; corrected p-value = 0.02; HR 2.6; 95%CI 1.35-4.89) as well as in the Durvalumab subgroup (p-value < 0.01; corrected p-value 0.047; HR 3.2; 95%CI 1.3-7.7).
|
Baseline characteristics |
||||||||
|
All (N = 174; events = 43) |
||||||||
|
Factor (# of cases) |
Univariate analysis |
Multivariate analysis |
||||||
|
P value |
HR |
95% CI |
P value/ BH adj. |
HR |
95% CI |
|||
|
Lower |
Upper |
Lower |
Upper |
|||||
|
Age (min: 36.39; median: 67.58; max: 90.73) |
0.388 |
0.987 |
0.959 |
1.016 |
0.37/0.37 |
0.986 |
0.96 |
1.02 |
|
Sex (female: 72; male: 102) |
0.623 |
1.141 |
0.675 |
1.928 |
0.23/0.37 |
1.6 |
0.76 |
3.18 |
|
Histology (nonSCC: 101; SCC: 73) |
<0.001 |
2.164 |
1.372 |
3.414 |
0.004/0.02 |
2.6 |
1.35 |
4.89 |
|
UICC (IIIa: 60; IIIbc: 114) |
0.534 |
1.135 |
0.761 |
1.694 |
0.29/0.37 |
1.4 |
0.73 |
2.85 |
|
MLR (low: 145; high: 29) |
0.044 |
2.084 |
1.021 |
4.253 |
0.11/0.28 |
1.8 |
0.8 |
3.7 |
|
Durvalumab (N = 105; events = 25) |
||||||||
|
Factor (# of cases) |
Univariate analysis |
Multivariate analysis |
||||||
|
P value |
HR |
95% CI |
P value / BH adj. |
HR |
95% CI |
|||
|
Lower |
Upper |
Lower |
Upper |
|||||
|
Age (min: 36.9; median: 67.7; max: 84.13) |
0.486 |
0.984 |
0.944 |
1.0266 |
0.74/0.74 |
1.4 |
0.95 |
1.04 |
|
Sex (female: 47; male: 58) |
0.449 |
0.724 |
0.315 |
1.667 |
0.52/0.65 |
0.99 |
0.54 |
3.4 |
|
Histology (nonSCC: 61; SCC: 44) |
<0.001 |
3.808 |
1.891 |
7.667 |
<0.01/0.047 |
3.2 |
1.3 |
7.7 |
|
UICC (IIIa: 37; IIIbc: 68) |
0.116 |
1.646 |
0.883 |
3.070 |
0.29/0.49 |
1.6 |
0.65 |
4 |
|
MLR (low: 88; high: 17) |
0.033 |
2.654 |
1.079 |
6.529 |
0.12/0.29 |
2 |
0.8 |
5 |
BH = Benjamini-Hochberg correction
Table 3b. Locoregional control (LRC): Multivariate analysis for MLR and treatment characteristics revealed MLR as the only significant parameter that impacted LRC in the whole cohort (p-value = 0.012; corrected p-value = 0.070; HR 0.38; 95%CI 0.18-0.81) and the Durvalumab subgroup (p-value < 0.05; corrected p-value = 0.15; HR 0.4; 95%CI 0.16-0.998), in which GTVLymphnodes was also significantly associated with better LRC (p-value = 0.01; corrected p-value = 0.07; HR = 1.0095; 95%CI 1.0022-1.017).
|
Treatment characteristics |
||||||||
|
All (N = 162; events = 42) |
||||||||
|
Factor (# of cases) |
Univariate analysis |
Multivariate analysis |
||||||
|
P value |
HR |
95% CI |
P value / BH adj. |
HR |
95% CI |
|||
|
Lower |
Upper |
Lower |
Upper |
|||||
|
GTV Tumour (min: 0.24; median: 48.4; mean: 87.08; max: 589.3) |
0.1 |
1.002 |
0.999 |
1.004 |
0.57/0.69 |
1.0008 |
0.998 |
1.0037 |
|
GTV Lymphnodes (min: 0; median: 29.2; mean: 56.65; max: 473) |
0.06 |
1.004 |
0.9999 |
1.008 |
0.15/0.25 |
1.0035 |
0.9987 |
1.0084 |
|
EQD2 Tumour (min: 24.8; median: 63.81; mean: 63.81; max: 100.0) |
0.06 |
0.977 |
0.95 |
1.001 |
0.17/0.25 |
0.9744 |
0.939 |
1.01 |
|
EQD2 Lymphnodes (min: 0; median: 57.29; mean: 48.92; max: 70.0) |
0.3 |
1.01 |
0.991 |
1.029 |
0.12/0.25 |
1.0195 |
0.9947 |
1.045 |
|
cCRT vs sCRT (cCRT: 38; sCRT: 124) |
0.5 |
0.78 |
0.38 |
1.6 |
0.84/0.84 |
1.08 |
0.51 |
2.28 |
|
MLR (high: 26; low: 136) |
0.01 |
0.42 |
0.21 |
0.86 |
0.012/0.07 |
0.38 |
0.18 |
0.81 |
|
Durvalumab (N = 103; events = 25) |
||||||||
|
Factor (# of cases) |
Univariate analysis |
Multivariate analysis |
||||||
|
P value |
HR |
95% CI |
P value / BH adj. |
HR |
95% CI |
|||
|
Lower |
Upper |
Lower |
Upper |
|||||
|
GTV Tumour (min: 0.24; median: 46; mean: 69.68; max: 483) |
0.7 |
1.001 |
0.997 |
1.005 |
0.62/0.62 |
1.0012 |
0.997 |
1.006 |
|
GTV Lymphnodes (min: 0; median: 26.7; mean: 45.89; max: 285) |
0.007 |
1.008 |
1.002 |
1.014 |
0.01/0.07 |
1.0095 |
1.0022 |
1.017 |
|
EQD2 Tumour (min: 32.5; median: 66; mean: 66.51; max: 100) |
0.3 |
0.978 |
0.94 |
1.022 |
0.4/0.5 |
0.977 |
0.923 |
1.03 |
|
EQD2 Lymphnodes (min: 0; median: 57.29; mean: 50.09; max: 70)) |
0.3 |
1.014 |
0.983 |
1.046 |
0.29/0.44 |
1.019 |
0.98 |
1.05 |
|
cCRT vs sCRT (cCRT: 23; sCRT: 80) |
0.9 |
1.033 |
0.387 |
2.755 |
0.2/0.4 |
2.032 |
0.685 |
6.03 |
|
MLR (high: 18; low: 85) |
0.08 |
0.46 |
0.19 |
1.113 |
<0.05/0.15 |
0.4 |
0.16 |
0.9983 |
BH = Benjamini-Hochberg correction
Table 3c. Locoregional control (LRC): Multivariate analysis for dMLR and clinical characteristics revealed histology as the only significant parameter that impacted LRC in the whole cohort (p-value <0.003; corrected p-value = 0.013; HR 2.75; 95%CI 1.4-5.3) and the Durvalumab subgroup (p-value = 0.007; corrected p-value = 0.03; HR 3.3; 95%CI 1.4-7.9).
|
Baseline characteristics |
||||||||
|
All (N = 172; events = 42) |
||||||||
|
Factor (# of cases) |
Univariate analysis |
Multivariate analysis |
||||||
|
P value |
HR |
95% CI |
P value / BH adj. |
HR |
95% CI |
|||
|
Lower |
Upper |
Lower |
Upper |
|||||
|
Age (min: 36.39; median: 67.36; max: 90.73) |
0.5 |
0.99 |
0.96 |
1.02 |
0.51/0.51 |
0.99 |
0.96 |
1.02 |
|
Sex (female: 69; male: 103) |
0.04 |
2.1 |
1.036 |
4.3 |
0.18/0.41 |
1.66 |
0.8 |
3.5 |
|
Histology (nonSCC: 100; SCC: 72) |
<0.001 |
3.1 |
1.66 |
5.9 |
<0.003/0.013 |
2.75 |
1.4 |
5.3 |
|
UICC (IIIa: 62; IIIbc: 110) |
0.3 |
1.47 |
0.75 |
2.9 |
0.3/0.4 |
1.4 |
0.7 |
2.8 |
|
dMLR (high: 26; low: 146) |
0.06 |
0.49 |
0.23 |
1.03 |
0.3/0.4 |
0.68 |
0.32 |
1.5 |
|
Durvalumab (N = 107; events = 25) |
||||||||
|
Factor (# of cases) |
Univariate analysis |
Multivariate analysis |
||||||
|
P value |
HR |
95% CI |
P value / BH adj. |
HR |
95% CI |
|||
|
Lower |
Upper |
Lower |
Upper |
|||||
|
Age (min: 36.39; median: 67.19; max: 84.13) |
0.44 |
1.004 |
0.96 |
1.045 |
0.86/0.86 |
0.996 |
0.96 |
1.03 |
|
Sex (female: 46; male: 61) |
0.65 |
1.8 |
0.75 |
4.3 |
0.49/0.72 |
1.37 |
0.56 |
3.39 |
|
Histology (nonSCC: 63; SCC: 44) |
0.001 |
3.6 |
1.6 |
8.4 |
<0.007/0.03 |
3.3 |
1.4 |
7.9 |
|
UICC (IIIa: 39; IIIbc: 68) |
0.2 |
1.7 |
0.7 |
4.1 |
0.28/0.7 |
1.6 |
0.66 |
4.1 |
|
dMLR (high: 18; low: 89) |
0.2 |
0.56 |
0.22 |
1.4 |
0.58/0.72 |
0.76 |
0.3 |
1.96 |
BH = Benjamini-Hochberg correction
Table 3d. Locoregional control (LRC): Multivariate analysis for dMLR and treatment characteristics revealed dMLR as the only significant parameter that impacted LRC in the whole cohort (p-value = 0.019; corrected p-value = 0.11; HR = 0.396; 95%CI 0.18-0.86).
|
Treatment characteristics |
||||||||
|
All (N = 160, events = 41) |
||||||||
|
Factor (# of cases) |
Univariate analysis |
Multivariate analysis |
||||||
|
P value |
HR |
95% CI |
P value / BH adj. |
HR |
95% CI |
|||
|
Lower |
Upper |
Lower |
Upper |
|||||
|
GTV Tumour (min: 0.24; median: 48; mean: 86.08; max: 589.3) |
0.1 |
1.002 |
0.999 |
1.004 |
0.6/0.73 |
1.0008 |
0.9978 |
1.0037 |
|
GTV Lymphnodes (min: 0; median: 27.65; mean: 49.77; max: 303.6) |
0.01 |
1.006 |
1.001 |
1.011 |
0.06/0.18 |
1.0055 |
0.9998 |
1.01 |
|
EQD2 Tumour (min: 24.8; median: 65; mean: 64.2; max: 100) |
0.05 |
0.976 |
0.95 |
0.9996 |
0.21/0.31 |
0.977 |
0.94 |
1.013 |
|
EQD2 Lymphnodes (min: 0; median: 57.29; mean: 49.2; max: 70) |
0.4 |
1.008 |
0.9899 |
1.027 |
0.16/0.31 |
1.018 |
0.99 |
1.043 |
|
cCRT vs sCRT (cCRT: 39; sCRT: 121) |
0.5 |
0.79 |
0.386 |
1.62 |
0.83/0.83 |
1.08 |
0.515 |
2.285 |
|
dMLR (high: 23; low: 137) |
0.03 |
0.455 |
0.213 |
0.957 |
0.019/0.11 |
0.396 |
0.18 |
0.86 |
|
Durvalumab (N = 105; events = 25) |
||||||||
|
Factor (# of cases) |
Univariate analysis |
Multivariate analysis |
||||||
|
P value |
HR |
95% CI |
P value / BH adj. |
HR |
95% CI |
|||
|
Lower |
Upper |
Lower |
Upper |
|||||
|
GTV Tumour (min: 0.24; median: 45; mean: 67; max: 483.8) |
0.6 |
1.001 |
0.997 |
1.005 |
0.59/0.59 |
1.0013 |
0.997 |
1.006 |
|
GTV Lymphnodes (min: 0; median: 24.6; mean: 45.04; max: 285) |
0.006 |
1.008 |
1.002 |
1.014 |
0.01/0.065 |
1.0094 |
1.0021 |
1.017 |
|
EQD2 Tumour (min: 32.5; median: 67.1; mean: 66.61; max: 100) |
0.3 |
0.976 |
0.93 |
1.02 |
0.38/0.46 |
0.975 |
0.92 |
1.03 |
|
EQD2 Lymphnodes (min: 0; median: 57.29; mean: 50.50; max: 70) |
0.3 |
1.013 |
0.986 |
1.041 |
0.27/0.4 |
1.02 |
0.99 |
1.056 |
|
cCRT vs sCRT (sCRT: 25; sCRT: 80) |
0.9 |
1.06 |
0.3976 |
2.83 |
0.2/0.4 |
2.01 |
0.683 |
5.93 |
|
dMLR (high: 17; low: 88) |
0.2 |
0.568 |
0.227 |
1.43 |
0.16/0.4 |
0.499 |
0.189 |
1.32 |
BH = Benjamini-Hochberg correction
Table 4a. Progression free survival (PFS): In the whole cohort multivariate analysis for MLR and clinical characteristics showed that UICC (p-value = 0.004; corrected p-value <0.01; HR 2.1; 95%CI 1.28-3.56) and MLR (p-value < 0.001; corrected p-value = 0.004; HR 0.43; 95%CI 0.26-0.7) significantly impacted PFS. This finding was corroborated in the Durvalumab subgroup with UICC (p-value = 0.04; corrected p-value = 0.073; HR 1.91; 95%CI 1.02-3.6) and MLR (p-value = 0.014; corrected p-value = 0.073; HR 0.4; 95%CI 0.22-0.85) as significant parameters. Additionally, histology was also significant (p-value = 0.044; corrected p-value = 0.073; HR 1.91; 95%CI 1.03-4.3).
|
Baseline characteristics |
||||||||
|
All (N = 174; events = 85) |
||||||||
|
Factor (# of cases) |
Univariate analysis |
Multivariate analysis |
||||||
|
P value |
HR |
95% CI |
P value/ BH adj. |
HR |
95% CI |
|||
|
Lower |
Upper |
Lower |
Upper |
|||||
|
Age (min: 36.39; median: 67.58; max: 90.73) |
0.3 |
0.99 |
0.959 |
1.016 |
0.12/0.17 |
0.98 |
0.96 |
1.005 |
|
Sex (female: 72; male: 102) |
1 |
0.99 |
0.65 |
1.5 |
0.48/0.48 |
0.85 |
0.54 |
1.34 |
|
Histology (nonSCC: 101; SCC: 73) |
0.1 |
45748 |
0.9 |
2.1 |
0.14/0.17 |
1.4 |
0.9 |
2.2 |
|
UICC (IIIa: 60; IIIbc: 114) |
<0.01 |
1.9 |
1.17 |
3.2 |
0.004/<0.01 |
2.1 |
1.28 |
3.56 |
|
MLR (high: 29; low: 145) |
<0.001 |
0.42 |
0.25 |
0.69 |
<0.001/0.004 |
0.43 |
0.26 |
0.7 |
|
Durvalumab (N = 105; events = 44) |
||||||||
|
Factor (# of cases) |
Univariate analysis |
Multivariate analysis |
||||||
|
P value |
HR |
95% CI |
P value / BH adj. |
HR |
95% CI |
|||
|
Lower |
Upper |
Lower |
Upper |
|||||
|
Age (min: 36.39; median 67.7; max: 84.13) |
0.8 |
1.005 |
0.97 |
1.037 |
0.59/0.59 |
0.991 |
0.96 |
1.02 |
|
Sex (female: 47; male: 58) |
0.9 |
1.05 |
0.58 |
1.91 |
0.59/0.59 |
0.84 |
0.44 |
1.6 |
|
Histology (nonSCC: 61; SCC: 44) |
0.04 |
1.8 |
1.02 |
3.34 |
0.044/0.073 |
1.91 |
1.02 |
3.6 |
|
UICC (IIIa: 37; IIIbc: 68) |
0.06 |
1.905 |
0.97 |
3.7 |
0.04/0.074 |
2.1 |
1.03 |
4.3 |
|
MLR (high: 17; low: 88) |
0.01 |
0.42 |
0.22 |
0.83 |
0.014/0.073 |
0.4 |
0.22 |
0.85 |
BH = Benjamini-Hochberg correction
Table 4b. Progression free survival (PFS): Multivariate analysis for MLR and treatment characteristics revealed EQD2Tumour (p-value <0.03; corrected p-value = 0.08; HR 0.97; 95%CI 0.95-0.9973) and MLR (p-value < 0.001; corrected p-value < 0.005; HR 0.39; 95%CI 0.22-0.68) with significant impact on PFS in the whole cohort. In the Durvalumab subgroup MLR was the only factor to remain significant (p-value = 0.0059; corrected p-value 0.035; HR 0.365; 95%CI 0.178-0.75).
|
Treatment characteristics |
||||||||
|
All (N = 162, events = 80) |
||||||||
|
Factor (# of cases) |
Univariate analysis |
Multivariate analysis |
||||||
|
P value |
HR |
95% CI |
P value / BH adj. |
HR |
95% CI |
|||
|
Lower |
Upper |
Lower |
Upper |
|||||
|
GTV Tumour (min: 0.24; median: 48.41; mean: 87.08; max: 589.3) |
0.1 |
1.001 |
0.9997 |
1.003 |
0.96/0.96 |
0.9999 |
0.9978 |
1.002 |
|
GTV Lymphnodes (min: 0; median: 29.2; mean: 56.65; max: 473) |
0.007 |
1.004 |
1.001 |
1.007 |
0.04/0.08 |
1.0035 |
1.0001 |
1.0068 |
|
EQD2 Tumour (min: 24.8; median: 63.81; mean: 63.81; max: 100) |
0.003 |
0.975 |
0.959 |
0.9915 |
<0.03/0.08 |
0.97 |
0.95 |
0.9973 |
|
EQD2 Lymphnodes (min: 0; median: 57.29; mean: 48.92; max: 70) |
0.6 |
1.003 |
0.9913 |
1.015 |
0.2/0.32 |
1.0092 |
0.99 |
1.024 |
|
cCRT vs sCRT (cCRT: 38; sCRT: 124) |
0.3 |
0.75 |
0.45 |
1.253 |
0.9/0.96 |
0.97 |
0.57 |
1.65 |
|
MLR (high: 26: low: 136) |
0.003 |
0.46 |
0.268 |
0.774 |
<0.001/<0.005 |
0.39 |
0.22 |
0.68 |
|
Durvalumab (N = 103; events = 43) |
||||||||
|
Factor (# of cases) |
Univariate analysis |
Multivariate analysis |
||||||
|
P value |
HR |
95% CI |
P value / BH adj. |
HR |
95% CI |
|||
|
Lower |
Upper |
Lower |
Upper |
|||||
|
GTV Tumour (min: 0.24; median: 46; mean: 69.68; max: 483.8) |
0.3 |
1.001 |
0.998 |
1.004 |
0.57/0.69 |
1.001 |
0.9975 |
1.0045 |
|
GTV Lymphnodes (min: 0; median: 26.7; mean: 45.89; max: 285) |
0.4 |
1.003 |
0.9969 |
1.008 |
0.1/0.2 |
1.005 |
0.9989 |
1.012 |
|
EQD2 Tumour (min: 32.5; median: 66; mean: 66.51; max: 100) |
0.2 |
0.98 |
0.945 |
1.011 |
0.23/0.35 |
0.9775 |
0.94 |
1.015 |
|
EQD2 Lymphnodes (min: 0; median: 57.29; mean: 50.09; max: 70) |
0.7 |
0.997 |
0.98 |
1.012 |
0.75/0.75 |
1.0029 |
0.985 |
1.021 |
|
cCRT vs sCRT (cCRT: 23; sCRT: 80) |
0.4 |
1.378 |
0.6 |
3.070 |
0.089/0.21 |
2.2 |
0.886 |
5.46 |
|
MLR (high: 18: low: 85) |
0.01 |
0.44 |
0.22 |
0.857 |
0.0059/0.035 |
0.365 |
0.178 |
0.75 |
BH = Benjamini-Hochberg correction
Table 4c. Progression free survival (PFS): Multivariate analysis for dMLR and clinical characteristics revealed that UICC (p-value < 0.009; corrected p-value = 0.023; HR 1.96; 95%CI 1.18-3.2) and dMLR (p-value = 0.04; corrected p-value = 0.02; HR 0.45; 95%CI 0.26-0.78) were significant factors to impact PFS in the whole cohort. This result was corroborated in the Durvalumab subgroup: UICC (p-value = 0.048; corrected p-value 0.12; HR 2.01; 95%CI 1.006-4.0) and dMLR (p-value = 0.018; corrected p-value = 0.088; HR 0.4; 95%CI 0.22-0.86).
|
Baseline characteristics |
||||||||
|
All (N = 172; events = 83) |
||||||||
|
Factor (# of cases) |
Univariate analysis |
Multivariate analysis |
||||||
|
P value |
HR |
95% CI |
P value / BH adj. |
HR |
95% CI |
|||
|
Lower |
Upper |
Lower |
Upper |
|||||
|
Age (min: 36.39; median: 67.36; max: 90.73) |
0.4 |
0.9905 |
0.97 |
1.013 |
0.13/0.22 |
0.98 |
0.96 |
1.005 |
|
Sex (female: 69; male: 103) |
0.21 |
1.001 |
0.64 |
1.56 |
0.53/0.53 |
0.86 |
0.55 |
1.36 |
|
Histology (nonSCC: 100; SCC: 72) |
0.1 |
1.4 |
0.9 |
2.16 |
0.31/0.39 |
1.27 |
0.8 |
2.02 |
|
UICC (IIIa: 62; IIIbc: 110) |
0.02 |
1.82 |
1.1 |
2.97 |
<0.009/0.023 |
1.96 |
1.18 |
3.2 |
|
dMLR (high: 26; low: 146) |
<0.001 |
0.42 |
0.25 |
0.7 |
0.04/0.02 |
0.45 |
0.26 |
0.78 |
|
Durvalumab (N = 107; events = 46) |
||||||||
|
Factor (# of cases) |
Univariate analysis |
Multivariate analysis |
||||||
|
P value |
HR |
95% CI |
P value / BH adj. |
HR |
95% CI |
|||
|
Lower |
Upper |
Lower |
Upper |
|||||
|
Age (min: 36.39; median: 67.19; max: 84.13) |
0.9 |
1.002 |
0.97 |
1.03 |
0.41/0.51 |
0.986 |
0.95 |
1.019 |
|
Sex (female: 46; male: 61) |
0.8 |
1.07 |
0.6 |
1.9 |
0.77/0.77 |
0.91 |
0.49 |
1.69 |
|
Histology (nonSCC: 63; SCC: 44) |
0.08 |
1.67 |
0.93 |
2.97 |
0.23/0.39 |
1.46 |
0.79 |
2.7 |
|
UICC (IIIa: 39; IIIbc: 68) |
0.06 |
1.86 |
0.97 |
3.55 |
0.048/0.12 |
2.01 |
1.006 |
4 |
|
dMLR (high: 18: low: 89) |
0.004 |
0.39 |
0.21 |
0.75 |
0.018/0.088 |
0.4 |
0.22 |
0.86 |
BH = Benjamini-Hochberg correction
Table 4d. Progression free survival (PFS): Multivariate analysis for dMLR and clinical characteristics revealed that dMLR was the only significant factor to impact PFS in the whole cohort (p-value = 0.002; corrected p-value = 0.014; HR 0.41; 95%CI: 0.2296-0.7264) as well as in the Durvalumab subgroup (p-value = 0.0017; corrected p-value = 0.01; HR 0.314; 95%CI 0.15-0.65).
|
Treatment characteristics |
||||||||
|
All (N = 160; events = 78) |
||||||||
|
Factor (# of cases) |
Univariate analysis |
Multivariate analysis |
||||||
|
P value |
HR |
95% CI |
P value / BH adj. |
HR |
95% CI |
|||
|
Lower |
Upper |
Lower |
Upper |
|||||
|
GTV Tumour (min: 0.24; median: 48; mean: 86.08; max: 589.3) |
0.1 |
1.001 |
0.9997 |
1.003 |
0.9/0.9 |
0.9999 |
0.998 |
1.002 |
|
GTV Lymphnodes (min: 0; median: 27.65; mean: 49.77; max: 303.6) |
0.3 |
1.002 |
0.998 |
1.006 |
0.7/0.84 |
1.0009 |
0.9964 |
1.0053 |
|
EQD2 Tumour (min: 24.8; median: 65; mean: 64.2; max: 100) |
0.01 |
0.978 |
0.96 |
0.995 |
0.19/0.058 |
0.97 |
0.947 |
0.995 |
|
EQD2 Lymphnodes (min: 0; median: 57.29; mean: 49.2; max: 70) |
0.4 |
1.005 |
0.993 |
1.017 |
0.096/0.19 |
1.0125 |
0.9978 |
1.027 |
|
cCRT vs sCRT (cCRT: 39; sCRT: 121) |
0.1 |
0.68 |
0.41 |
1.123 |
0.54/0.81 |
0.847 |
0.4977 |
1.441 |
|
dMLR (high: 23; low: 137) |
0.003 |
0.44 |
0.257 |
0.762 |
0.002/0.014 |
0.41 |
0.2296 |
0.7264 |
|
Durvalumab (N = 105; events = 45) |
||||||||
|
Factor (# of cases) |
Univariate analysis |
Multivariate analysis |
||||||
|
P value |
HR |
95% CI |
P value / BH adj. |
HR |
95% CI |
|||
|
Lower |
Upper |
Lower |
Upper |
|||||
|
GTV Tumour (min: 0.24; median: 45; mean: 67.5; max: 483.8) |
0.4 |
1.001 |
0.998 |
1.004 |
0.97/0.97 |
1.0001 |
0.9967 |
1.0035 |
|
GTV Lymphnodes (min: 0; median: 24.6; mean: 45.04; max: 285) |
0.5 |
1.002 |
0.996 |
1.008 |
0.25/0.45 |
1.0037 |
0.997 |
1.01 |
|
EQD2 Tumour (min: 32.5; median: 67.1; mean: 66.61; max: 100) |
0.2 |
0.977 |
0.946 |
1.01 |
0.21/0.45 |
0.976 |
0.9395 |
1.014 |
|
EQD2 Lymphnodes (min: 0; median: 57.29; mean: 50.5; max: 70) |
0.9 |
0.9989 |
0.984 |
1.014 |
0.39/0.48 |
1.0078 |
0.9899 |
1.0259 |
|
cCRT vs sCRT (cCRT: 25: sCRT: 80) |
0.9 |
1.032 |
0.496 |
2.146 |
0.3/0.48 |
1.5434 |
0.68 |
3.5 |
|
dMLR (high: 17: low: 88) |
0.003 |
0.3818 |
0.199 |
0.7322 |
0.0017/0.01 |
0.314 |
0.15 |
0.65 |
BH = Benjamini-Hochberg correction
Manuscript text, methods section 2.3
The Benjamini-Hochberg method was used to adjust for multiple testing.
Manuscript text, results section 3.4
When MLR was tested in the univariate (UVA) and multivariate model (MVA) together with baseline and treatment characteristics, the following variables had a significant impact on LRC: histology (p-value = 0.004; corrected p-value = 0.02; HR 2.6; 95%CI 1.35-4.89; table 3a) and MLR (p-value = 0.012; corrected p-value = 0.070; HR 0.38; 95%CI 0.18-0.81; table 3b). This finding could be corroborated in the Durvalumab subgroup (histology: p-value < 0.01; corrected p-value 0.047; HR 3.2; 95%CI 1.3-7.7; table 3a; MLR: p-value < 0.05; corrected p-value = 0.15; HR 0.4; 95%CI 0.16-0.998, table 3b). Additionally, GTVTumour was also a significant factor with respect to LRC in the Durvalumab cohort (p-value = 0.01; corrected p-value = 0.07; HR = 1.0095; 95%CI 1.0022-1.017). A separate analysis with the same baseline and treatment characteristics combined with dMLR instead of MLR revealed that histology was the only predictive parameter for LRC both in the whole cohort (p-value <0.003; corrected p-value = 0.013; HR 2.75; 95%CI 1.4-5.3; table 3c) and the Durvalumab subgroup (p-value = 0.007; corrected p-value = 0.03; HR 3.3; 95%CI 1.4-7.9; table 3c). Furthermore, dMLR had an impact on LRC in the whole cohort (p-value = 0.019; corrected p-value = 0.11; HR = 0.396; 95%CI 0.18-0.86; table 3d).
(…)
When MLR was tested in the MVA model together with baseline and treatment characteristics, the following variables had a significant impact on PFS: UICC (p-value = 0.004; corrected p-value <0.01; HR 2.1; 95%CI 1.28-3.56; table 4a), MLR (table 4a: p-value < 0.001; corrected p-value = 0.004; HR 0.43; 95%CI 0.26-0.7; table 4b: p-value < 0.001; corrected p-value < 0.005; HR 0.39; 95%CI 0.22-0.68) and EQD2Tumour (p-value <0.03; corrected p-value = 0.08; HR 0.97; 95%CI 0.95-0.9973; table 4b). In the Durvalumab subgroup, risk factors for a PFS were histology (p-value = 0.044; corrected p-value = 0.073; HR 1.91; 95%CI 1.03-4.3; table 4a), UICC (p-value = 0.04; corrected p-value = 0.073; HR 1.91; 95%CI 1.02-3.6; table 4a) and MLR (table 4a: p-value = 0.014; corrected p-value = 0.073; HR 0.4; 95%CI 0.22-0.85; table 4b: p-value = 0.0059; corrected p-value 0.035; HR 0.365; 95%CI 0.178-0.75). When repeating the same analysis with dMLR instead of MLR, UICC (p-value < 0.009; corrected p-value = 0.023; HR 1.96; 95%CI 1.18-3.2; table 4c) and dMLR (table 4c: p-value = 0.04; corrected p-value = 0.02; HR 0.45; 95%CI 0.26-0.78; table 4d: p-value = 0.002; corrected p-value = 0.014; HR 0.41; 95%CI: 0.2296-0.7264) were significant factors to be associated with PFS in the whole cohort. This result was also found in the Durvalumab subgroup: UICC (p-value = 0.048; corrected p-value 0.12; HR 2.01; 95%CI 1.006-4.0, table 4c) and dMLR (table 4c: p-value = 0.018; corrected p-value = 0.088; HR 0.4; 95%CI 0.22-0.86; table 4d: p-value = 0.0017; corrected p-value = 0.01; HR 0.314; 95%CI 0.15-0.65).
